

# Fermionization and boundary states in 1+1 dimensions

Yoshiki Fukusumi[1], Yuji Tachikawa[2] and Yunqin Zheng[2,3]

**1** Department of Physics, Faculty of Science,
University of Zagreb, HR-10000 Zagreb, Croatia
**2** Kavli Institute for the Physics and Mathematics of the Universe,
University of Tokyo, Kashiwa, Chiba 277-8583, Japan
**3** Institute for Solid State Physics,
University of Tokyo, Kashiwa, Chiba 277-8581, Japan

## Abstract

In the last few years it was realized that every fermionic theory in 1+1 dimensions is a generalized Jordan-Wigner transform of a bosonic theory with a non-anomalous $\mathbb{Z}_2$ symmetry. In this note we determine how the boundary states are mapped under this correspondence. We also interpret this mapping as the fusion of the original boundary with the fermionization interface.

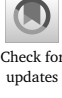
# 1 Introduction and Summary

It is a classic result that the Jordan-Wigner transformation [1] allows us to map any 1-dimensional spin chain written in terms of $\sigma^{(i)}_{X,Y,Z}$ preserving a $\mathbb{Z}_2$ symmetry generated by $\prod \sigma^{(i)}_{Z}$ in terms of fermion operators $\psi^{(i)}$, $\bar{\psi}^{(i)}$. One of the most famous applications of this transformation is the solution of the Ising model by mapping it into a free fermionic chain [2]. This transformation was so effective that in the older literature the fermionic model and the bosonic model related in this manner were not carefully distinguished. It is to be stressed, however, that the models related in this manner do differ, and have different energy spectrum once boundary conditions are carefully taken account.

Therefore, there are much that could be gained by maintaining this distinction. For example, we now know that an abstract version of the Jordan-Wigner transformation can be formulated in the continuum, and any continuum fermionic theory in 1+1 dimensions is obtained from a bosonic (1+1)-dimensional system with a non-anomalous $\mathbb{Z}_2$ symmetry by this transformation [3–8], generalizing the relation between the Ising conformal field theory (CFT) and the Majorana fermion CFT. It was also learned recently[1] that fermionic versions of unitary minimal models can be constructed in this manner [11–13].[2][3] The aim of this paper is to study how this map between fermionic models and bosonic models with $\mathbb{Z}_2$ symmetry affects the boundary states.

Let us first recall the basic facts of the continuum version of the Jordan-Wigner transformation. We start from a bosonic theory A with a non-anomalous $\mathbb{Z}_2$ global symmetry. Another bosonic theory D can be introduced by

$$\mathsf{D} := \mathsf{A}/\mathbb{Z}_2 \,, \tag{1.1}$$

where dividing by $\mathbb{Z}_2$ stands for orbifolding, or equivalently gauging of the $\mathbb{Z}_2$ symmetry. This theory D is known to possess a dual $\mathbb{Z}_2$ symmetry [21] and gauging it reproduces the original theory: $\mathsf{A} = \mathsf{D}/\mathbb{Z}_2$. The fermionic theory F is obtained with the help of another theory Kitaev, the low-energy limit of the topologically nontrivial phase of Kitaev's Majorana chain [22].[4] The formula is

$$\mathsf{F} := (\mathsf{A} \times \mathsf{Kitaev})/\mathbb{Z}_2 \,, \tag{1.2}$$

---

[1]Note added in proof: The authors learned very recently that fermionic minimal models had already been found in 1988 [9, 10]. They thank Prof. V. Petkova for information.

[2]There should also be a generalization from $\mathbb{Z}_2$ to $\mathbb{Z}_N$, where $\mathbb{Z}_N$ symmetric critical points are related to $\mathbb{Z}_N$ parafermionic models. Historically, such generalization was first suggested in [14] in 1985, where it was conjectured that the $m$-multicritical points of $\mathbb{Z}_N$ Fateev-Zamolodchikov model should be described by the second parafermionic models with parameter $m$ for $N \geq 3$ in [14]; the corresponding bosonic coset models were later given in [15, 16]. From this perspective, it was equally natural to expect that the $m$-multicritical Ising model should also have a fermionic counterpart, whose explicit lattice construction was given only very recently in [12]. It would be interesting to revisit and develop these points further. See e.g. [17–19] for recent works.

[3]See also [20] for a recent study of fermionic RCFTs with small number of irreducible representations.

[4]This theory is also known as the Arf theory, since its partition function on a closed two-dimensional surface with spin structure is given by its Arf invariant.

where the orbifold is with respect to the diagonal $\mathbb{Z}_2$ symmetry. We can also define a closely related theory F′ starting from D

$$\mathsf{F}' := (\mathsf{D} \times \mathsf{Kitaev})/\mathbb{Z}_2 \,, \tag{1.3}$$

which satisfies[5]

$$\mathsf{F}' = \mathsf{F} \times \mathsf{Kitaev} \,. \tag{1.4}$$

We place these theories on a circle. The bosonic theories can be put on a circle without or with $\mathbb{Z}_2$ twist. Similarly, the fermionic theories can be put on the Neveu-Schwarz (NS) sector, or equivalently the antiperiodic sector, and the Ramond (R) sector, or equivalently the periodic sector. The Hilbert spaces of theories A, D, F, F′ are known to decompose in the following manner:

| A | untwisted | twisted | | D | untwisted | twisted |
|---|---|---|---|---|---|---|
| even | $S$ | $U$ | | even | $S$ | $T$ |
| odd | $T$ | $V$ | | odd | $U$ | $V$ |
| F | NS | R | | F′ | NS | R |
| $(-1)^F = +1$ | $S$ | $U$ | | $(-1)^F = +1$ | $S$ | $T$ |
| $(-1)^F = -1$ | $V$ | $T$ | | $(-1)^F = -1$ | $V$ | $U$ |

$$(1.5)$$

We denote the Hilbert spaces in the respective sectors by $\mathcal{H}_{S,T,U,V}$, so that the untwisted and twisted Hilbert spaces of the theory A have the decomposition

$$\mathcal{H}^{\mathsf{A}} = \mathcal{H}_S \oplus \mathcal{H}_T \,, \tag{1.6}$$

$$\mathcal{H}^{\mathsf{A}}_{\mathrm{tw}} = \mathcal{H}_U \oplus \mathcal{H}_V \,, \tag{1.7}$$

for example. Since F and F′ differ only by a decoupled $\mathbb{Z}_2^F$ SPT, this makes it tricky to distinguish them in the lattice realizations on closed chains.

We now consider boundary conditions of the bosonic theory A, which can be broadly classified into two types by their behavior under the $\mathbb{Z}_2$ symmetry. Namely, there are $\mathbb{Z}_2$ invariant ones, which we denote by $i, j, \ldots$, and the ones which break the $\mathbb{Z}_2$ symmetry, which therefore form pairs we can denote by $a\pm, b\pm, \ldots$. We assume the theory is defined on a cylinder of size $L_1 \times L_2$, where $L_1$ is the length of the open spatial direction and $L_2$ is the circumference of the closed temporal direction. We denote the Hilbert space on a finite interval with boundary conditions $\alpha$ on the left boundary and $\beta$ on the right boundary as $\mathcal{H}_{\alpha|\beta}$. The boundary states are obtained by 90 degree rotation of the spacetime, hence the temporal direction becomes open, and the boundary states are defined on the initial and final slices. We denote the corresponding boundary states as $|i\rangle^{\mathsf{A}}$, $|a\pm\rangle^{\mathsf{A}}$, etc. Note that a $\mathbb{Z}_2$ invariant boundary condition $\alpha$ can be placed on a circle which is twisted by the $\mathbb{Z}_2$ operation, which defines twisted boundary states $|i\rangle^{\mathsf{A}}_{\mathrm{tw}}$ taking values in the twisted Hilbert space. The (twisted) boundary conditions and the (twisted) boundary states are related by the Cardy condition,

$$\langle \alpha | e^{-L_1 H_{\mathrm{closed}}} | \beta \rangle = \mathrm{Tr}_{\mathcal{H}_{\alpha|\beta}} e^{-L_2 H_{\mathrm{open}}} \,, \qquad _{\mathrm{tw}}\langle \alpha | e^{-L_1 H_{\mathrm{closed}}} | \beta \rangle_{\mathrm{tw}} = \mathrm{Tr}_{\mathcal{H}_{\alpha|\beta}} g\, e^{-L_2 H_{\mathrm{open}}} \,, \tag{1.8}$$

---

[5]In terms of the partition function coupled to $\mathbb{Z}_2$ background field $B$ and spin structure $\rho$, we have $Z_{\mathsf{F}}(\rho) = 2^{-g} \sum_B Z_{\mathsf{A}}(B)(-1)^{\mathrm{Arf}[\rho+B]}$. Similarly, we have $Z_{\mathsf{F}'}(\rho) = 2^{-g} \sum_B Z_{\mathsf{A}}(B)(-1)^{q_\rho[B]}$ where $q_\rho[B] \equiv \mathrm{Arf}[\rho+B] - \mathrm{Arf}[\rho]$ is the quadratic form associated to the spin structure. These two are related as follows: $Z_{\mathsf{F}'}[\rho] = Z_{\mathsf{F}}[\rho](-1)^{\mathrm{Arf}[\rho]}$. In the works [6,12] by one of the authors (YT) and other collaborators, F is defined to be the fermionization of A, whereas in most of the other works of this topic e.g. [3,5,8,23], F′ is defined to be the fermionization of A instead. We note that when A is the completely trivial theory, the theory F is trivial while F′ is equal to Kitaev.

where $g$ is the $\mathbb{Z}_2$ action on $\mathcal{H}_{\alpha|\beta}$. The Cardy condition (1.8) constrains the normalization of the boundary states.

Let us further consider the boundary conditions and boundary states of a fermionic theory F. The discussion is similar to the bosonic theory as the previous paragraph. The global symmetry is the fermion parity symmetry $\mathbb{Z}_2^F$. The boundary conditions can be classified to be $\mathbb{Z}_2^F$ even and odd ones. The fermion along the closed time cycle can have either anti-periodic (NS) spin structure or periodic (R) spin structure. For the R spin structure, there is a $\mathbb{Z}_2^F$ line inserted along the spatial direction, which is analogous to the twisted boundary condition in the bosonic case. In summary, the boundary conditions are denoted by $\alpha$. After 90 degree rotation, the boundary conditions are mapped to boundary states, which we denote as $|\alpha\rangle_{\mathrm{NS}}$ or $|\alpha\rangle_{\mathrm{R}}$. The boundary states and boundary conditions satisfy the *spin Cardy condition*:

$$\mathstrut_{\mathrm{NS}}\langle\alpha|e^{-L_1 H_{\mathrm{closed}}}|\beta\rangle_{\mathrm{NS}} = \mathrm{Tr}_{\mathcal{H}_{\alpha|\beta}}\,e^{-L_2 H_{\mathrm{open}}}\,, \qquad \mathstrut_{\mathrm{R}}\langle\alpha|e^{-L_1 H_{\mathrm{closed}}}|\beta\rangle_{\mathrm{R}} = \mathrm{Tr}_{\mathcal{H}_{\alpha|\beta}}(-1)^F e^{-L_2 H_{\mathrm{open}}}\,. \quad (1.9)$$

In the following sections, we do not specify whether Hamiltonian is defined on the "open" or "closed" string, because it should be evident from the discussion.

Our main result is the expressions of the boundary states of the theories D, F and F$'$, which will be derived in detail later. Here we will simply summarize them. In the $\mathbb{Z}_2$-orbifold theory D, the $\mathbb{Z}_2$-invariant boundary condition $i$ of the original theory A splits into two types $i\pm$, which are exchanged under the emergent $\widehat{\mathbb{Z}}_2$ symmetry. Conversely, the pair of $\mathbb{Z}_2$-breaking boundary conditions $a\pm$ is combined into a single $\mathbb{Z}_2$-invariant condition $a$. Their boundary states are given as follows:

$$|i+\rangle^{\mathsf{D}} = \frac{1}{\sqrt{2}}(|i\rangle^{\mathsf{A}} + |i\rangle^{\mathsf{A}}_{\mathrm{tw}}), \qquad |i-\rangle^{\mathsf{D}} = \frac{1}{\sqrt{2}}(|i\rangle^{\mathsf{A}} - |i\rangle^{\mathsf{A}}_{\mathrm{tw}}),$$
$$|a\rangle^{\mathsf{D}} = \frac{1}{\sqrt{2}}(|a+\rangle^{\mathsf{A}} + |a-\rangle^{\mathsf{A}}), \qquad |a\rangle^{\mathsf{D}}_{\mathrm{tw}} = \frac{1}{\sqrt{2}}(|a+\rangle^{\mathsf{A}} - |a-\rangle^{\mathsf{A}}). \tag{1.10}$$

Here $\widehat{\mathbb{Z}}_2$ exchanges $|i+\rangle^{\mathsf{D}}$ and $|i-\rangle^{\mathsf{D}}$, while $|a\rangle^{\mathsf{D}}$ and $|a\rangle^{\mathsf{D}}_{\mathrm{tw}}$ are both $\widehat{\mathbb{Z}}_2$ even. The boundary states $|i\pm\rangle^{\mathsf{D}}$, $|a\rangle^{\mathsf{D}}$ and $|a\rangle^{\mathsf{D}}_{\mathrm{tw}}$ satisfy Cardy's condition (1.8).

In the fermionic theories, boundary conditions can be classified into two types, those with and without an unpaired Majorana fermion zero mode. In particular, Kitaev itself has such an unpaired Majorana fermion zero mode [24]. Let us start from a $\mathbb{Z}_2$-invariant boundary condition $i$ of the theory A. When stacked with Kitaev, this boundary has a Majorana fermion $\psi$, but the $\mathbb{Z}_2$ quotient removes it since this fermion is $\mathbb{Z}_2$ odd. Conversely, when we start from a $\mathbb{Z}_2$-breaking boundary condition $a\pm$, this additional $\mathbb{Z}_2$-odd Majorana fermion $\psi$ is kept since the $\mathbb{Z}_2$ gauge symmetry is broken here, resulting in a boundary with a Majorana fermion. We denote the resulting boundary states as $|i\rangle^{\mathsf{F}}$ and $|a\psi\rangle^{\mathsf{F}}$, etc. The corresponding boundary states are given as follows:

$$|i\rangle^{\mathsf{F}}_{\mathrm{NS}} = |i\rangle^{\mathsf{A}}\,, \qquad\qquad |i\rangle^{\mathsf{F}}_{\mathrm{R}} = |i\rangle^{\mathsf{A}}_{\mathrm{tw}}\,,$$
$$|a\psi\rangle^{\mathsf{F}}_{\mathrm{NS}} = |a+\rangle^{\mathsf{A}} + |a-\rangle^{\mathsf{A}}\,, \qquad |a\psi\rangle^{\mathsf{F}}_{\mathrm{R}} = |a+\rangle^{\mathsf{A}} - |a-\rangle^{\mathsf{A}}\,. \tag{1.11}$$

Here $|a\psi\rangle^{\mathsf{F}}_{\mathrm{R}}$ has $(-1)^F = -1$ while the other three have $(-1)^F = +1$. We note that the boundary state in the R-sector of a boundary condition with an unpaired Majorana fermion vanishes due to the Majorana fermion zero mode coming from the periodicity. Our $|a\psi\rangle^{\mathsf{F}}_{\mathrm{R}}$ is defined with an insertion of the Majorana fermion operator to absorb this zero mode. We do not repeat

this comment unless absolutely necessary. The boundary states $|i\rangle^F_{NS/R}$ and $|a\psi\rangle^F_{NS/R}$ satisfy the spin Cardy condition (1.9).

In the theory $F'$, the assignment is reversed, and we find $|i\psi\rangle^{F'}$ and $|a\rangle^{F'}$. These boundary states are given as follows:

$$|i\psi\rangle^{F'}_{NS} = \sqrt{2}\,|i\rangle^A, \qquad\qquad |i\psi\rangle^{F'}_{R} = \sqrt{2}\,|i\rangle^A_{tw},$$
$$|a\rangle^{F'}_{NS} = \frac{1}{\sqrt{2}}(|a+\rangle^A + |a-\rangle^A), \qquad |a\rangle^{F'}_{R} = \frac{1}{\sqrt{2}}(|a+\rangle^A - |a-\rangle^A). \qquad (1.12)$$

Again $|i\psi\rangle^F_R$ has $(-1)^F = -1$ while the other three have $(-1)^F = +1$. Note that the theory F and $F'$ have rather different sets of boundary conditions if $A \neq D$, even though the NS sectors of F and $F'$ are the same and the R sectors of F and $F'$ only differ in their assignments of the fermion parity. The boundary states $|i\psi\rangle^F_{NS/R}$ and $|a\rangle^F_{NS/R}$ satisfy the spin Cardy condition (1.9).

We find it also useful to present our results in a different way, showing explicitly to which sector $\mathcal{H}_{S,T,U,V}$ the boundary states belong. We first organize the boundary states of theory A into four blocks[6] (1.5):

| A | untwisted | twisted |
|---|---|---|
| even | $\|i\rangle_S = \|i\rangle^A$ <br> $\|a\rangle_S = \frac{1}{\sqrt{2}}\left(\|a+\rangle^A + \|a-\rangle^A\right)$ | $\|i\rangle_U = \|i\rangle^A_{tw}$ |
| odd | $\|a\rangle_T = \frac{1}{\sqrt{2}}\left(\|a+\rangle^A - \|a-\rangle^A\right)$ | |

$$(1.13)$$

Then the boundary states of theory D can be presented as:

| D | untwisted | twisted |
|---|---|---|
| even | $\|a\rangle_S = \|a\rangle^D$ <br> $\|i\rangle_S = \frac{1}{\sqrt{2}}\left(\|i+\rangle^D + \|i-\rangle^D\right)$ | $\|a\rangle_T = \|a\rangle^D_{tw}$ |
| odd | $\|i\rangle_U = \frac{1}{\sqrt{2}}\left(\|i+\rangle^D - \|i-\rangle^D\right)$ | |

$$(1.14)$$

For the theory F and $F'$ they are given by

| F | NS | R |
|---|---|---|
| $(-1)^F = +1$ | $\|i\rangle_S = \|i\rangle^F_{NS}$ <br> $\sqrt{2}\|a\rangle_S = \|a\psi\rangle^F_{NS}$ | $\|i\rangle_U = \|i\rangle^F_R$ |
| $(-1)^F = -1$ | | $\sqrt{2}\|a\rangle_T = \|a\psi\rangle^F_R$ |

$$(1.15)$$

and

| $F'$ | NS | R |
|---|---|---|
| $(-1)^F = +1$ | $\|a\rangle_S = \|a\rangle^{F'}_{NS}$ <br> $\sqrt{2}\|i\rangle_S = \|i\psi\rangle^{F'}_{NS}$ | $\|a\rangle_T = \|a\rangle^{F'}_R$ |
| $(-1)^F = -1$ | | $\sqrt{2}\|i\rangle_U = \|i\psi\rangle^{F'}_R$ |

$$(1.16)$$

We see that the states are exchanged as in (1.5).

The rest of the note is organized as follows. In Sec. 2, we provide a detailed derivation of the expressions of the boundary states of the theories D, F and $F'$ we listed above. We will

---

[6]Note that $|a\rangle_S$ and $|a\rangle_T$ are not boundary states of A. They are $\mathbb{Z}_2$ even and odd linear combinations of boundary states in the untwisted sector. Likewise, $|i\rangle_S$ and $|i\rangle_U$ are not boundary states of D, but instead are linear combinations of them.

not use the property that the theories are conformal. Instead, we will only use the fact that the theory on a torus can be consistently interpreted with either direction as the Euclidean time direction. We then provide in Sec. 3 various examples. Finally in Sec. 4 we interpret our findings in terms of interfaces between our theories A, D, F and F′. We also see that when the original theory A has a duality interface implementing the $\mathbb{Z}_2$ orbifold, so that A ≃ D, the fermionized theory F ≃ F′ has an anomalous $\mathbb{Z}_2$ symmetry. We have an appendix A collecting various facts on diagonal invariants of rational conformal field theories (RCFTs) which can be used to construct their fermionic versions and fermionic boundary conditions.

Before proceeding, the authors note that a paper with a large overlap [25] appeared when this work was being completed, in which the fermionic boundary states of fermionic minimal models and the effect of anomalous chiral $\mathbb{Z}_2$ symmetry were studied in detail; he also carefully discusses the boundary states of exceptional fermionic minimal models introduced in [13]. The authors also note that another paper with a significant overlap [26] appeared on the arXiv on the same day as this one. The authors thank the authors of [26] for discussions and for coordinating the submission.

# 2 Mapping of the boundary states

In this section we study how the boundary states change under gauging and fermionization. The objective is to give a detailed derivation of the mappings we already announced in Sec. 1.

## 2.1 Gauging $\mathbb{Z}_2$: A ↔ D

Let us start with a bosonic theory A with $\mathbb{Z}_2$ global symmetry. We demand that $\mathbb{Z}_2$ is non-anomalous, so that there is no obstruction to gauge it. We denote the resulting $\mathbb{Z}_2$ gauged theory by D, whose emergent $\mathbb{Z}_2$ symmetry we denote by $\widehat{\mathbb{Z}}_2$. We study how the boundary states of theory A transform under gauging. The content of this section is essentially known in the vast existing literature on the boundary conditions of $\mathbb{Z}_2$ orbifolds, but we find it useful to discuss in detail as a preparation of our discussion of the fermionization.

**Setup:** We start with the general definition of gauging, and will first assume that the spacetime is a torus without boundary. Denote the partition function of theory A as $Z_{\mathsf{A}}^{(r,s)}$ where $r, s \in \{0, 1\}$ count the number of $\mathbb{Z}_2$ defect lines in the spatial and temporal directions. We use the symbols $Z_{\mathsf{D}}^{(m,n)}$ similarly for the gauged theory D, where $m, n$ now count the number of defect lines for the emergent $\widehat{\mathbb{Z}}_2$ global symmetry. We then have the relation[7]

$$Z_{\mathsf{D}}^{(m,n)} = \frac{1}{2} \sum_{(r,s)} (-1)^{rn-ms} Z_{\mathsf{A}}^{(r,s)}. \tag{2.1}$$

Next, we demand that the spacetime has nontrivial boundaries in the temporal direction, i.e. time direction is open. We place two boundary states $|\alpha\rangle$ and $|\beta\rangle$ in the initial time and the final time respectively, and denote the partition function as $Z_{\mathsf{A}}^{(r,s)}[\langle\beta|\alpha\rangle]$. As discussed in the introduction, there are broadly speaking two types of boundaries: we denote the $\mathbb{Z}_2$-invariant

---

[7]More generally, the partition functions of A and D are related as $Z_{\mathsf{D}}[\widehat{B}] = \frac{1}{2^g} \sum_B Z_{\mathsf{A}}[B](-1)^{\int B \cup \widehat{B}}$, where $B$ and $\widehat{B}$ are the $\mathbb{Z}_2$ and dual $\mathbb{Z}_2$ background fields in A and D respectively. The pair $(r,s)$ in (2.1) labels the nontrivial holonomies of the background field $\int_x B = s$, $\int_y B = r$ mod 2 where the integrals are along the non-contractible cycles in the direction $x$ and $y$, respectively. The pair $(m,n)$ is obtained from $\widehat{B}$ in a similar manner.

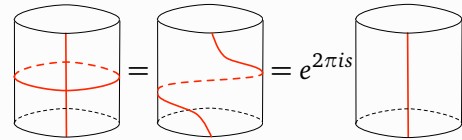

Figure 1: $\mathbb{Z}_2$ defect, $\mathbb{Z}_2$ charge and the spin of the twisted Hilbert space. The $\mathbb{Z}_2$ charge of a state in the $\mathbb{Z}_2$ is given by the leftmost figure, which can be deformed to the middle figure. Therefore the charge is related to the spin of the state. When we put a $\mathbb{Z}_2$-invariant boundary condition on the top and bottom slices, we can always unwind the defect, and the spin is always integer valued.

ones by $|i\rangle^A$, $|j\rangle^A$, ... and $\mathbb{Z}_2$-breaking pairs by $|a\pm\rangle^A$, $|b\pm\rangle^A$, .... We also denote the twisted sector boundary states by $|i\rangle^A_{tw}$, $|j\rangle^A_{tw}$ .... We also employ analogous notations for theory D.

$\mathbb{Z}_2$ **charge of** $|i\rangle^A$: Since the boundary state $|i\rangle^A$ and $|i\rangle^A_{tw}$ are invariant under $\mathbb{Z}_2$, we need to find their $\mathbb{Z}_2$ charge. The boundary state $|i\rangle^A$ in the untwisted Hilbert space is $\mathbb{Z}_2$ even rather than $\mathbb{Z}_2$ odd. To see this, we use Cardy's analysis to transform the partition function to the open string channel. Consider the theory on a cylinder of size $L_1 \times L_2$ where $L_1$ is along the open direction and $L_2$ is along the closed direction. Suppose $|i\rangle^A$ has $\mathbb{Z}_2$ charge $s = \pm 1$, thus $g|i\rangle^A = s|i\rangle^A$. Then

$$Z_A^{(1,0)}[^A\langle j|i\rangle^A] = {}^A\langle j|e^{-L_1 H}g|i\rangle^A = s\,{}^A\langle j|e^{-L_1 H}|i\rangle^A = s\,\text{Tr}_{\mathcal{H}_{j|i}}e^{-L_2 H}. \tag{2.2}$$

We also have

$$Z_A^{(1,0)}[^A\langle j|i\rangle^A] = \text{Tr}_{\mathcal{H}^g_{j|i}}e^{-L_2 H}, \tag{2.3}$$

where $\mathcal{H}^g_{j|i}$ is the defect Hilbert space in the open string channel with two boundary conditions labeled by $j$ and $i$ with an explicit insertion of $\mathbb{Z}_2$ line in the middle. Now, $\text{Tr}_{\mathcal{H}}e^{-L_2 H}$ for any $\mathcal{H}$ should be a sum of exponentials with non-negative coefficients, therefore we need to have $s = +1$.[8] Summarizing, we see $g|i\rangle^A$ is $+|i\rangle^A$ instead of $-|i\rangle^A$, where $g$ is the $\mathbb{Z}_2$ generator.

$\mathbb{Z}_2$ **charge of** $|i\rangle^A_{tw}$: To find the $\mathbb{Z}_2$ charge of $|i\rangle^A_{tw}$, we make use of the following trick [27,28]. By definition, the $\mathbb{Z}_2$ charge of a state in the twisted Hilbert space can be found by considering the configuration of the $\mathbb{Z}_2$ line operators in the leftmost figure of Fig. 1. Assuming that the $\mathbb{Z}_2$ symmetry is non-anomalous, this configuration can be continuously deformed to a line operator winding the cylinder once, therefore the $\mathbb{Z}_2$-charge equals $e^{2\pi i s}$ where $s$ is the spin of the state $|\phi\rangle \in \mathcal{H}_{tw}$:

$$g|\phi\rangle = e^{2\pi i s}|\phi\rangle, \tag{2.4}$$

where $g$ is the $\mathbb{Z}_2$ generator. When the state in question is the boundary state for a $\mathbb{Z}_2$-invariant boundary condition, we can always unwind $g$, say at the top time slice, to make it straight, hence $e^{2\pi i s} = 1$. In summary, both the states $|i\rangle^A$, $|i\rangle^A_{tw}$ are $\mathbb{Z}_2$ even.

**Rough properties of the boundary states:** We first look for the boundary states in theory D that is $\mathbb{Z}_2$-invariant under the emergent $\widehat{\mathbb{Z}}_2$ symmetry. Let us symbolically denote it as $|\alpha\rangle$ and

---

[8]The authors thank Shu-Heng Shao for suggesting this argument.

place it in the initial time slice. We also put a reference state $|R\rangle$ in the final time slice. (The choice of reference state does not affect our discussion). We have the following relations

$$
\begin{aligned}
Z_D^{(0,0)}[\langle R|\alpha\rangle] &= Z_D^{(1,0)}[\langle R|\alpha\rangle], \\
Z_D^{(0,1)}[\langle R|\alpha\rangle] &= Z_D^{(1,1)}[\langle R|\alpha\rangle] = 0.
\end{aligned}
\tag{2.5}
$$

The first equality follows from $|\alpha\rangle$ being $\widehat{\mathbb{Z}}_2$ symmetric, where we can move the $\widehat{\mathbb{Z}}_2$ line along the spatial direction down to the initial time slice and let it be absorbed by the initial state $|\alpha\rangle$. The second and third equations follow from the fact that $|\alpha\rangle$ do not live in the twisted sector. Combining (2.1) and (2.5), we have

$$
\begin{aligned}
Z_A^{(0,0)}[\langle R|\alpha\rangle] &= Z_A^{(1,0)}[\langle R|\alpha\rangle], \\
Z_A^{(0,1)}[\langle R|\alpha\rangle] &= Z_A^{(1,1)}[\langle R|\alpha\rangle] = 0.
\end{aligned}
\tag{2.6}
$$

The relations (2.6) imply that $|\alpha\rangle$ can not be $|i\rangle_{tw}^A$ because otherwise $Z_A^{(0,0)}[\langle R|\alpha\rangle] = Z_A^{(1,0)}[\langle R|\alpha\rangle] = 0$, thus $Z_A^{(r,s)}[\langle R|\alpha\rangle] = 0$ for all $r,s$, which is absurd. The first equality in (2.6) also implies that $|\alpha\rangle$ can not be proportional to $|a+\rangle^A - |a-\rangle^A$ neither, because the relations (2.6) imply that $|\alpha\rangle$ should have $\mathbb{Z}_2$ eigenvalue $+1$. Therefore $|\alpha\rangle$ can only be proportional to $|i\rangle^A$ or $|a+\rangle^A + |a-\rangle^A$.

We further look for the symmetric boundary state $|\beta\rangle$ in the twisted sector of theory D. Applying an analysis that was carried out in the last paragraph, we find

$$
\begin{aligned}
Z_D^{(0,1)}[\langle R|\beta\rangle] &= Z_D^{(1,1)}[\langle R|\beta\rangle], \\
Z_D^{(0,0)}[\langle R|\beta\rangle] &= Z_D^{(1,0)}[\langle R|\beta\rangle] = 0.
\end{aligned}
\tag{2.7}
$$

Combining with (2.1), we have

$$
\begin{aligned}
Z_A^{(0,0)}[\langle R|\beta\rangle] &= -Z_A^{(1,0)}[\langle R|\beta\rangle], \\
Z_A^{(0,1)}[\langle R|\beta\rangle] &= Z_A^{(1,1)}[\langle R|\beta\rangle] = 0.
\end{aligned}
\tag{2.8}
$$

This implies that $|\beta\rangle$ can only be proportional to $|a+\rangle^A - |a-\rangle^A$. This in fact would suggest that $|\alpha\rangle$ should be proportional to $|a+\rangle^A + |a-\rangle^A$, but not $|i\rangle^A$. Hence we will denote

$$
\begin{aligned}
|a\rangle^D &\propto |a+\rangle^A + |a-\rangle^A, \\
|a\rangle_{tw}^D &\propto |a+\rangle^A - |a-\rangle^A.
\end{aligned}
\tag{2.9}
$$

We further look for the $\mathbb{Z}_2$-odd boundary state $|\gamma\rangle$ in the untwisted sector of theory D. This means that

$$
\begin{aligned}
Z_D^{(0,0)}[\langle R|\gamma\rangle] &= -Z_D^{(1,0)}[\langle R|\gamma\rangle], \\
Z_D^{(0,1)}[\langle R|\gamma\rangle] &= Z_D^{(1,1)}[\langle R|\gamma\rangle] = 0.
\end{aligned}
\tag{2.10}
$$

Combining with (2.1), we have

$$
\begin{aligned}
Z_A^{(0,1)}[\langle R|\gamma\rangle] &= Z_A^{(1,1)}[\langle R|\gamma\rangle], \\
Z_A^{(0,0)}[\langle R|\gamma\rangle] &= Z_A^{(1,0)}[\langle R|\gamma\rangle] = 0.
\end{aligned}
\tag{2.11}
$$

Hence $|\gamma\rangle$ should be the twisted sector state of theory A, $|i\rangle_{tw}^A$, which indeed is $\mathbb{Z}_2$ even.

Lastly we look for the $\mathbb{Z}_2$-odd boundary state $|\eta\rangle$ in the twisted sector of D. This means that

$$
\begin{aligned}
Z_D^{(0,1)}[\langle R|\eta\rangle] &= -Z_D^{(1,1)}[\langle R|\eta\rangle], \\
Z_D^{(0,0)}[\langle R|\eta\rangle] &= Z_D^{(1,0)}[\langle R|\eta\rangle] = 0.
\end{aligned}
\tag{2.12}
$$

Combining with (2.1), we have

$$
\begin{aligned}
Z_A^{(0,1)}[\langle R|\eta\rangle] &= -Z_A^{(1,1)}[\langle R|\eta\rangle], \\
Z_A^{(0,0)}[\langle R|\eta\rangle] &= Z_A^{(1,0)}[\langle R|\eta\rangle] = 0.
\end{aligned}
\tag{2.13}
$$

Because there is no $\mathbb{Z}_2$ odd state in the twisted sector of theory A, there is no solution to (2.13). Hence all the boundary states in theory D that are related to $|i\rangle^A$ and $|i\rangle_{tw}^A$ do not live in the twisted sector of $\widehat{\mathbb{Z}}_2$. This suggests that there are $|i+\rangle^D$ and $|i-\rangle^D$ which are mapped to each other under $\widehat{\mathbb{Z}}_2$, which satisfy

$$
\begin{aligned}
|i+\rangle^D + |i-\rangle^D &\propto |i\rangle^A, \\
|i+\rangle^D - |i-\rangle^D &\propto |i\rangle_{tw}^A.
\end{aligned}
\tag{2.14}
$$

**Fixing the normalization constants:** To determine the coefficients in (2.9) and (2.14), we need to make use of Cardy's conditions. We consider a theory on a cylinder of size $L_1 \times L_2$, with two boundary conditions $\alpha$ and $\beta$. We then have Cardy's conditions

$$
\langle \alpha | e^{-L_1 H} | \beta \rangle = \text{Tr}_{\mathcal{H}_{\alpha|\beta}} e^{-L_2 H}, \qquad {}_{tw}\langle \alpha | e^{-L_1 H} | \beta \rangle_{tw} = \text{Tr}_{\mathcal{H}_{\alpha|\beta}} g e^{-L_2 H},
\tag{2.15}
$$

where $g$ is the $\mathbb{Z}_2$ generator. Below, we will drop $e^{-L_1 H}$ and $e^{-L_2 H}$ for brevity.

Let us first consider (2.14). Suppose $|i+\rangle^D + |i-\rangle^D = P|i\rangle^A$ and $|i+\rangle^D - |i-\rangle^D = Q|i\rangle_{tw}^A$. Then the relations (2.15) imply that

$$
\begin{aligned}
{}^A\langle i|j\rangle^A &= \text{Tr}_{\mathcal{H}_{i|j}^+} + \text{Tr}_{\mathcal{H}_{i|j}^-}, \\
{}_{tw}^A\langle i|j\rangle_{tw}^A &= \text{Tr}_{\mathcal{H}_{i|j}^+} - \text{Tr}_{\mathcal{H}_{i|j}^-},
\end{aligned}
\tag{2.16}
$$

where $\mathcal{H}_{i|j}^\pm$ is the part of the open Hilbert space with boundary conditions $i$, $j$ placed on two ends, with the $\mathbb{Z}_2$ charge $\pm 1$. Hence

$$
\begin{aligned}
{}^D\langle i\pm|j\pm\rangle^D &= \frac{1}{4}(|P|^2 + |Q|^2)\text{Tr}_{\mathcal{H}_{i|j}^+} + \frac{1}{4}(|P|^2 - |Q|^2)\text{Tr}_{\mathcal{H}_{i|j}^-}, \\
{}^D\langle i\pm|j\mp\rangle^D &= \frac{1}{4}(|P|^2 - |Q|^2)\text{Tr}_{\mathcal{H}_{i|j}^+} + \frac{1}{4}(|P|^2 + |Q|^2)\text{Tr}_{\mathcal{H}_{i|j}^-}.
\end{aligned}
\tag{2.17}
$$

For $|i\pm\rangle^D$ to be legal boundary states, we should have $\frac{1}{4}(|P|^2 \pm |Q|^2)$ to be non-negative integers. It is then reasonable to set $P = Q = \sqrt{2}$, which is the minimal solution of $P$ and $Q$. We thus get

$$
|i\pm\rangle^D = \frac{1}{\sqrt{2}}(|i\rangle^A \pm |i\rangle_{tw}^A).
\tag{2.18}
$$

Since gauging twice goes back to the original theory itself, we also finds the normalization in (2.9)

$$
|a\rangle^D = \frac{1}{\sqrt{2}}(|a+\rangle^A + |a-\rangle^A), \qquad |a\rangle_{tw}^D = \frac{1}{\sqrt{2}}(|a+\rangle^A - |a-\rangle^A).
\tag{2.19}
$$

These results were presented in a slightly different manner in Sec. 1.

## 2.2 Fermionizing $\mathbb{Z}_2$: A $\to$ F, F′

**Setup:** Let us further consider fermionizing the $\mathbb{Z}_2$ global symmetry of A, and denote the resulting theory as F. More precisely, we first stack the theory A with a Kitaev chain in the nontrivial phase, and gauge the diagonal $\mathbb{Z}_2$ global symmetry: F = (A × Kitaev)/$\mathbb{Z}_2$. The partition functions on the torus are related as follows:[9]

$$Z_F^{(m,n)} = \frac{1}{2}\sum_{(r,s)}(-1)^{(m+r)(n+s)}Z_A^{(r,s)}, \tag{2.20}$$

where the pair $(m,n)$ now specifies the periodicity of the spin structure on the torus, in the convention where $m, n = 0$ is the NS sector and $m, n = 1$ is the R sector.

We first need to recall the argument we already gave in the introduction: consider a $\mathbb{Z}_2$-invariant boundary condition $i$ of the theory A. When stacked with Kitaev, this boundary has a Majorana fermion $\psi$, which is removed by the $\mathbb{Z}_2$ quotient since it is $\mathbb{Z}_2$ odd. In contrast, when we start from a $\mathbb{Z}_2$-breaking boundary condition $a\pm$, this $\mathbb{Z}_2$-odd Majorana fermion $\psi$ is retained since the $\mathbb{Z}_2$ gauge symmetry is broken here, resulting in a boundary with a Majorana fermion. Furthermore, acting by $\psi$ essentially converts $a+$ to $a-$ and vice versa. Therefore we find two types of boundary conditions: $|i\rangle^F$ without Majorana fermion which comes from a $\mathbb{Z}_2$-invariant boundary condition $i$ of theory A, and $|a\psi\rangle^F$ with a Majorana fermion which comes from a pair of $\mathbb{Z}_2$-breaking boundary conditions $a\pm$ of theory A.

**Rough properties of the boundary conditions:** The way we determine the boundary states is analogous to what we did in the case A $\leftrightarrow$ D studied above. When there are nontrivial boundaries along the temporal direction, we again put a reference state $|R\rangle$ in the final time slice, and put a boundary state $|\alpha\rangle$ in the initial slice which we will study. Now we need to consider four choices: NS/R sectors and $(-1)^F = \pm 1$, which we study in turn.

Let us first look into the state $|\alpha\rangle$ in the NS sector with fermion parity $(-1)^F = 1$. We have

$$Z_F^{(0,0)}[\langle R|\alpha\rangle] = Z_F^{(1,0)}[\langle R|\alpha\rangle],$$
$$Z_F^{(0,1)}[\langle R|\alpha\rangle] = Z_F^{(1,1)}[\langle R|\alpha\rangle] = 0. \tag{2.21}$$

The first equality follows from $(-1)^F = 1$ and the last two equalities are because $|\alpha\rangle$ is in the NS sector. Combining with (2.20), we have

$$Z_A^{(0,0)}[\langle R|\alpha\rangle] = Z_A^{(1,0)}[\langle R|\alpha\rangle],$$
$$Z_A^{(0,1)}[\langle R|\alpha\rangle] = Z_A^{(1,1)}[\langle R|\alpha\rangle] = 0, \tag{2.22}$$

which implies that

$$|\alpha\rangle \propto |i\rangle^A \quad\text{or}\quad |a+\rangle^A + |a-\rangle^A. \tag{2.23}$$

Let us further look into the state $|\beta\rangle$ in the R sector with $(-1)^F = 1$. We have

$$Z_F^{(0,1)}[\langle R|\beta\rangle] = Z_F^{(1,1)}[\langle R|\beta\rangle],$$
$$Z_F^{(0,0)}[\langle R|\beta\rangle] = Z_F^{(1,0)}[\langle R|\beta\rangle] = 0. \tag{2.24}$$

---

[9]See the footnote 5 for the expression on a general surface of genus $g$.

Combining with (2.20), we have

$$
\begin{aligned}
Z_A^{(0,1)}[\langle R|\beta\rangle] &= Z_A^{(1,1)}[\langle R|\beta\rangle], \\
Z_A^{(0,0)}[\langle R|\beta\rangle] &= Z_A^{(1,0)}[\langle R|\beta\rangle] = 0.
\end{aligned}
\tag{2.25}
$$

The only solution $|\beta\rangle$ to (2.25) is $|i\rangle_{\mathrm{tw}}^A$. We will denote $|\beta\rangle$ as $|i\rangle_R^F$, i.e.

$$
|i\rangle_R^F \propto |i\rangle_{\mathrm{tw}}^A.
\tag{2.26}
$$

Let us further look into the boundary state $|\gamma\rangle$ in the R sector with $(-1)^F = -1$. This means

$$
\begin{aligned}
Z_F^{(0,1)}[\langle R|\gamma\rangle] &= -Z_F^{(1,1)}[\langle R|\gamma\rangle], \\
Z_F^{(0,0)}[\langle R|\gamma\rangle] &= Z_F^{(1,0)}[\langle R|\gamma\rangle] = 0.
\end{aligned}
\tag{2.27}
$$

Combining with (2.20), we have

$$
\begin{aligned}
Z_A^{(0,0)}[\langle R|\gamma\rangle] &= -Z_A^{(1,0)}[\langle R|\gamma\rangle], \\
Z_A^{(0,1)}[\langle R|\gamma\rangle] &= Z_A^{(1,1)}[\langle R|\gamma\rangle] = 0.
\end{aligned}
\tag{2.28}
$$

The only solution $|\gamma\rangle$ to (2.28) is $|a+\rangle^A - |a-\rangle^A$. We will denote $|\gamma\rangle$ as $|a\psi\rangle_R^F$, thus

$$
|a\psi\rangle_R^F \propto |a+\rangle^A - |a-\rangle^A.
\tag{2.29}
$$

Finally let us look into the boundary state $|\eta\rangle$ in the NS sector with $(-1)^F = -1$. This means

$$
\begin{aligned}
Z_F^{(0,0)}[\langle R|\eta\rangle] &= -Z_F^{(1,0)}[\langle R|\eta\rangle], \\
Z_F^{(0,1)}[\langle R|\eta\rangle] &= Z_F^{(1,1)}[\langle R|\eta\rangle] = 0.
\end{aligned}
\tag{2.30}
$$

Combining with (2.20), we have

$$
\begin{aligned}
Z_A^{(0,1)}[\langle R|\eta\rangle] &= -Z_A^{(1,1)}[\langle R|\eta\rangle], \\
Z_A^{(0,0)}[\langle R|\eta\rangle] &= Z_A^{(1,0)}[\langle R|\eta\rangle] = 0.
\end{aligned}
\tag{2.31}
$$

We note, however, that there is no boundary state in the twisted sector of theory A which is $\mathbb{Z}_2$ odd. Hence there is no boundary state in NS sector with $(-1)^F = -1$ in theory F.

To encode the $(-1)^F$ quantum number to the boundary state, we introduce $|\pm\rangle$, representing $(-1)^F = \pm 1$ respectively. In summary, we find

$$
\begin{aligned}
|i\rangle_{\mathrm{NS}}^F &\propto |i\rangle^A \otimes |+\rangle, \\
|a\psi\rangle_{\mathrm{NS}}^F &\propto (|a+\rangle^A + |a-\rangle^A) \otimes |+\rangle, \\
|i\rangle_R^F &\propto |i\rangle_{\mathrm{tw}}^A \otimes |+\rangle, \\
|a\psi\rangle_R^F &\propto (|a+\rangle^A - |a-\rangle^A) \otimes |-\rangle.
\end{aligned}
\tag{2.32}
$$

where we remind the reader that our $|a\psi\rangle_R^F$ includes an insertion of a Majorana fermion to absorb the zero mode, which is required to make the state nonzero.



**Fixing the normalization constants:** We need to further determine the normalization constants in (2.32). Suppose the normalization constants are $P_1, P_2, P_3, P_4$ for the four relations in (2.32) respectively. Exchanging the time directions, we find

$$\phantom{}_{\mathsf{NS}}^{\mathsf{F}}\langle i|j\rangle_{\mathsf{NS}}^{\mathsf{F}} = |P_1|^2(\mathrm{Tr}_{\mathcal{H}_{i|j}^+} + \mathrm{Tr}_{\mathcal{H}_{i|j}^-}), \tag{2.33}$$

$$\phantom{}_{\mathsf{NS}}^{\mathsf{F}}\langle a\psi|b\psi\rangle_{\mathsf{NS}}^{\mathsf{F}} = |P_2|^2(\mathrm{Tr}_{\mathcal{H}_{a+|b+}} + \mathrm{Tr}_{\mathcal{H}_{a-|b-}} + \mathrm{Tr}_{\mathcal{H}_{a+|b-}} + \mathrm{Tr}_{\mathcal{H}_{a-|b+}}) \tag{2.34}$$

$$= 2|P_2|^2(\mathrm{Tr}_{\mathcal{H}_{a+|b+}} + \mathrm{Tr}_{\mathcal{H}_{a+|b-}}), \tag{2.35}$$

$$\phantom{}_{\mathsf{R}}^{\mathsf{F}}\langle i|j\rangle_{\mathsf{R}}^{\mathsf{F}} = |P_3|^2(\mathrm{Tr}_{\mathcal{H}_{i|j}^+} - \mathrm{Tr}_{\mathcal{H}_{i|j}^-}), \tag{2.36}$$

$$\phantom{}_{\mathsf{R}}^{\mathsf{F}}\langle a\psi|b\psi\rangle_{\mathsf{R}}^{\mathsf{F}} = |P_4|^2(\mathrm{Tr}_{\mathcal{H}_{a+|b+}} + \mathrm{Tr}_{\mathcal{H}_{a-|b-}} - \mathrm{Tr}_{\mathcal{H}_{a+|b-}} - \mathrm{Tr}_{\mathcal{H}_{a-|b+}}) \tag{2.37}$$

$$= 2|P_4|^2(\mathrm{Tr}_{\mathcal{H}_{a+|b+}} - \mathrm{Tr}_{\mathcal{H}_{a+|b-}}), \tag{2.38}$$

$$\phantom{}_{\mathsf{NS}}^{\mathsf{F}}\langle i|a\psi\rangle_{\mathsf{NS}}^{\mathsf{F}} = P_1^* P_2(\mathrm{Tr}_{\mathcal{H}_{i|a+}} + \mathrm{Tr}_{\mathcal{H}_{i|a-}}), \tag{2.39}$$

$$\phantom{}_{\mathsf{R}}^{\mathsf{F}}\langle i|a\psi\rangle_{\mathsf{R}}^{\mathsf{F}} = P_3^* P_4(\mathrm{Tr}_{\mathcal{H}_{i|a+}} - \mathrm{Tr}_{\mathcal{H}_{i|a-}}), \tag{2.40}$$

where we used the fact that the $\mathbb{Z}_2$ symmetry of the A-theory implies that $\mathrm{Tr}_{\mathcal{H}_{a+|b+}} = \mathrm{Tr}_{\mathcal{H}_{a-|b-}}$ and $\mathrm{Tr}_{\mathcal{H}_{a+|b-}} = \mathrm{Tr}_{\mathcal{H}_{a-|b+}}$ to convert (2.34), (2.37) to (2.35), (2.38), respectively.

The boundary states of F should satisfy the spin Cardy conditions (1.9). This requires that the coefficients on the right hand side should be non-negative integers for NS states, while integers for R states.[10] There are two minimal[11] solutions of $P_i$'s,

$$P_1 = P_2 = P_3 = P_4 = 1 \tag{2.41}$$

and

$$P_1 = P_3 = \sqrt{2}, \quad P_2 = P_4 = \frac{1}{\sqrt{2}}, \tag{2.42}$$

which yield minimal integer coefficients on the right hand sides of (2.33)-(2.39), hence satisfy the spin Cardy conditions. The correct choice for the theory F is (2.41). To see this, it suffices to consider the case when A is the completely trivial theory. There is a single $\mathbb{Z}_2$-symmetric boundary condition $i$, for which $\mathcal{H}_{i|i} = 1$ is one dimensional and $\mathbb{Z}_2$-even. In this theory F $= (\mathsf{A} \times \mathsf{Kitaev})/\mathbb{Z}_2$ is also a trivial theory, which has a trivial boundary, for which $\phantom{}_{\mathsf{NS}}^{\mathsf{F}}\langle i|i\rangle_{\mathsf{NS}}^{\mathsf{F}} = 1$. This fixes $P_1 = 1$.

For more general theories, the equations (2.33) and (2.36) simply state the fact that for $\mathbb{Z}_2$-invariant boundary condition $i, j$ of theory A, the corresponding boundary conditions of the theory F is obtained by attaching a Majorana fermion $\psi$ and simply removing it by the $\mathbb{Z}_2$ projection. Therefore the open Hilbert space in the fermionic theory with boundary conditions $i$ and $j$ is simply equal to the Hilbert space of the original theory $\mathcal{H}_{i|j}$, and the fermion parity $(-1)^F$ equals the original $\mathbb{Z}_2$-charge $U$.

To interpret (2.35) and (2.38), we note that each of these boundary conditions hosts a single Majorana zero mode. Therefore, with two boundaries, we need to quantize a pair of such modes, providing a factor of 2. We note that the relative sign in (2.38) does not correspond to $(-1)^F$, since in our definition we inserted a Majorana fermion operator on the boundaries $|a\psi\rangle^{\mathsf{F}}$ and $|b\psi\rangle^{\mathsf{F}}$.

---

[10] Here we choose the convention that the coefficient $P_3^* P_4$ in (2.40) is positive. One can alternatively let $P_3^* P_4$ be negative. The two conventions are related by flipping the sign of the boundary Majorana zero mode inserted to make $|a\psi\rangle_{\mathsf{R}}^{\mathsf{F}}$ nonzero, and neither sign of the Majorana zero mode is privileged.

[11] Minimal means to keep the coefficients on the right hand side of (2.33)-(2.39) to be minimal non-negative integers.

In summary, we found[12]

$$|i\rangle_{NS}^{F} = |i\rangle^{A} \otimes |+\rangle \,,$$
$$|a\psi\rangle_{NS}^{F} = (|a+\rangle^{A} + |a-\rangle^{A}) \otimes |+\rangle \,,$$
$$|i\rangle_{R}^{F} = |i\rangle_{tw}^{A} \otimes |+\rangle \,,$$
$$|a\psi\rangle_{R}^{F} = (|a+\rangle^{A} - |a-\rangle^{A}) \otimes |-\rangle \,. \tag{2.43}$$

We can further append a nontrivial Kitaev chain to F to obtain F′. As the relation between F and A is equal to that of the relation of F′ and D, we can easily find from (2.18) and (2.19) that

$$|i\psi\rangle_{NS}^{F'} = \sqrt{2}\,|i\rangle^{A} \otimes |+\rangle \,,$$
$$|a\rangle_{NS}^{F'} = \frac{1}{\sqrt{2}}(|a+\rangle^{A} + |a-\rangle^{A}) \otimes |+\rangle \,,$$
$$|i\psi\rangle_{R}^{F'} = \sqrt{2}\,|i\rangle_{tw}^{A} \otimes |-\rangle \,,$$
$$|a\rangle_{R}^{F'} = \frac{1}{\sqrt{2}}(|a+\rangle^{A} - |a-\rangle^{A}) \otimes |+\rangle \,. \tag{2.44}$$

As a consistency check, attaching a Kitaev chain amounts to change the number of Majorana zero modes in the R sector states, hence the amplitude of the boundary states are exchanged between 1 and $\sqrt{2}$, exactly as shown in (2.44) and (2.43). Our final results (2.43) and (2.44) were already presented in Sec. 1 as (1.11) and (1.12). There, we suppressed $|\pm\rangle$, while just remarking their fermion parity explicitly.

## 2.3 Comments

Before proceeding, we would like to make two remarks.

**On two common normalizations:** Recall that during the derivation, we demanded that the overlap of any pair of boundary states has a consistent Hamiltonian interpretation in the open channel to deduce the proportionality coefficients. This is different from the standard viewpoint in hep-th in the following sense.

Consider a massless free Majorana fermion in 1+1d, which allows two boundary conditions $\psi_L = \pm\psi_R$ on the boundary. In hep-th, it is often stated that if we choose a wrong pair of boundary conditions $\alpha$, $\beta$ on the two ends of a segment, there is a bulk fermionic Majorana zero mode, which makes the system not quantizable. This problem can be cured by putting a Majorana fermion by hand. In such a case, it is $\sqrt{2}\langle\alpha|\beta\rangle$ which has a sensible Hamiltonian interpretation in the open channel. The recent paper [25] uses this normalization.

Instead, in this paper we chose to use a viewpoint which is more in line with cond-mat, or which happens when we realize the Majorana fermion on the lattice for example. In this case, if $\psi_L = +\psi_R$ is a boundary condition without a boundary Majorana fermion, then the boundary condition $\psi_L = -\psi_R$ comes with a boundary Majorana zero mode. In this case any pair of boundary conditions is quantizable, but then we need to introduce additional normalization constants of $\sqrt{2}^{\pm 1}$ in front of the boundary states. Then the boundary states for a single Majorana fermion with boundary conditions $\psi_L = +\psi_R$ and $\psi_L = -\psi_R$ would have

---

[12]As one of the authors emphasized in [29], the linear combination of Cardy states appears naturally as zero mode of fermionic model [30,31]. This type of states has also captured attention in the condensed matter physics community [32–34].

different norms. This $\sqrt{2}$ factor can also be observed as the $\sqrt{2}$ fold degeneracy of Majorana chain [29–31].

**Relation with the lattice Jordan-Wigner transformations:** Recall also that during the derivation, we saw the relation between the open Hilbert spaces of the original bosonic model A and the transformed fermionic model F, which we gave in (2.33) to (2.39). We also saw that $\mathbb{Z}_2$-invariant boundary conditions give rise to boundary conditions without localized Majorana fermions, and that $\mathbb{Z}_2$-breaking boundary conditions to boundary conditions with localized Majorana fermions.

Here we show that this is also what we have when we perform the Jordan-Wigner transformation on an open chain. In this sense, our transformation form A to F is indeed the Jordan-Wigner transformation in disguise.

Let us first consider the case of standard $\mathbb{Z}_2$-invariant boundary conditions on open chains. Namely, we consider an open chain of $L$ sites with the operators $\sigma_{X,Y,Z}^{(s)}$, ($s = 1, \ldots, L$). We take a Hamiltonian $H$ commuting with the $\mathbb{Z}_2$ symmetry generated by $\prod_s \sigma_Z^{(s)}$. The boundary conditions at both ends are $\mathbb{Z}_2$-invariant. Such a Hamiltonian can be rewritten in terms of Majorana fermions $\psi^{(t)}$, ($t = 1, \ldots, 2L$) using the standard formulas. Then almost by definition, the Hamiltonian written in the fermionic variables is equal to the Hamiltonian written in terms of $\sigma_{X,Y,Z}^{(s)}$, and the fermion parity is equal to $\prod_s \sigma_Z^{(s)}$. This reproduces the relation (2.33) and (2.36).

Next, we consider a particular class of $\mathbb{Z}_2$-breaking boundary conditions on an open chain, and see it reproduces (2.34). The existence of localized Majorana zero modes is also clearly visible.

We again start from the same Hamiltonian $\mathbb{Z}_2$-invariant $H$ on a chain of $L$ sites, and express it in terms of $\sigma_{X,Z}^{(s)}$ without $\sigma_Y^{(s)}$. At the leftmost site $i = 1$, we drop the terms from $H$ involving $\sigma_Z^{(1)}$, and also perform an analogous operation on the other end $i = L$. Let us denote the resulting Hamiltonian by $H'$. Such a Hamiltonian $H'$ commutes with $\sigma_X^{(1)}$ and $\sigma_X^{(L)}$, and therefore $H'$ and $\sigma_X^{(1,L)}$ can be simultaneously diagonalized. In other words, $H'$ splits into four sectors $H'_{\pm\pm}$ depending on the eigenvalues of $\sigma_X^{(1,L)}$, and the four sectors describe $\mathbb{Z}_2$-breaking boundary conditions imposed on both ends. Now we perform the standard Jordan-Wigner transformation for $H'$, which is still $\mathbb{Z}_2$-symmetric. The construction guarantees that $H'$ does not involve $\psi^{(1)}$ and $\psi^{(2L)}$, meaning that they commute with $H'$. We found that each end has an unpaired Majorana zero mode.

To make this abstract discussion more concrete, take for example, the standard critical Ising chain

$$-H_{\text{Ising}} = \sum_{s=1}^{L} \sigma_Z^{(s)} + \sum_{s=1}^{L-1} \sigma_X^{(s)} \sigma_X^{(s+1)}. \tag{2.45}$$

The modified chain has the Hamiltonian

$$-H'_{\text{Ising}} = \sum_{s=2}^{L-1} \sigma_Z^{(s)} + \sum_{s=1}^{L-1} \sigma_X^{(s)} \sigma_X^{(s+1)}, \tag{2.46}$$

which splits into four sectors described by

$$-H'_{\text{Ising}\pm\pm} = (\sum_{s=2}^{L-1} \sigma_Z^{(s)}) \pm \sigma_X^{(2)} + (\sum_{s=2}^{L-2} \sigma_X^{(s)} \sigma_X^{(s+1)}) \pm \sigma_X^{(L-1)}, \tag{2.47}$$

acting on sites $s = 2, \ldots, L-1$. This $H'_{\text{Ising}\pm\pm}$ indeed breaks $\mathbb{Z}_2$ at both boundaries and has the standard forms there.

Now $H'$ written in the Majorana fermion variables is simply

$$-H' = \sum_{t=2}^{2L-2} i\psi^{(t)}\psi^{(t+1)}, \tag{2.48}$$

and commutes with $\psi^{(1)}$ and $\psi^{(2L)}$. We indeed found localized Majorana zero modes, and also reproduced the relation (2.34).

# 3 Examples

In section 2, we discussed the transformation of boundary states under gauging and fermionizing the $\mathbb{Z}_2$ global symmetry on general grounds. In this section, we apply the general results in section 2 to concrete examples. We often use standard results in the theory of RCFTs, such as Ishibashi and Cardy states [35, 36]; for general aspects of BCFT, see e.g. [37–39]. We refer the readers to Appendix A for a brief review and further references.

## 3.1 Fermionic $SU(2)_k$ WZW model

Let us first discuss the fermionic version of the $SU(2)_k$ Wess-Zumino-Witten (WZW) model. Recall that $SU(2)_k$ WZW model has $k+1$ primaries labeled by $j = 0, \frac{1}{2}, \cdots, \frac{k}{2}$ with $\Delta_j = \frac{j(j+1)}{k+2}$. The fusion rule is

$$[j] \otimes [j'] = [|j - j'|] \oplus [|j - j'| + 1] \oplus \cdots \oplus [\min(j + j', k - j - j')]. \tag{3.1}$$

The modular $S$ matrix is

$$S_{jj'} = \sqrt{\frac{2}{k+2}} \sin\frac{\pi(2j+1)(2j'+1)}{k+2}. \tag{3.2}$$

There is a $\mathbb{Z}_2$ global symmetry, generated by the Verlinde line $\mathcal{L}_{\frac{k}{2}}$ associated with the primary operator $[\frac{k}{2}]$. This is non-anomalous when $k$ is even and anomalous when $k$ is odd. In the following we only consider the case $k$ is even.

For a given $j$, the conformal primary and its descendants $|j, N\rangle$ all have the same $\mathbb{Z}_2$ eigenvalue $\frac{S_{\frac{k}{2}j}}{S_{0j}} = (-1)^{2j}$. The Ishibashi states are schematically given by

$$|j\rangle\!\rangle \sim \sum_N |j, N\rangle \otimes |\bar{j}, \bar{N}\rangle , \tag{3.3}$$

which are $\mathbb{Z}_2$ eigenstates with eigenvalue $(-1)^{2j}$, i.e. $g|j\rangle\!\rangle = (-1)^{2j}|j\rangle\!\rangle$. Therefore $\mathbb{Z}_2$ acts on the Cardy state as

$$g|j\rangle = |\tfrac{k}{2} - j\rangle . \tag{3.4}$$

The only $\mathbb{Z}_2$-invariant Cardy state is $|\frac{k}{4}\rangle$.

The twisted Cardy state $|\frac{k}{4}\rangle_{\text{tw}}$ can be determined using the general formula given in Appendix A. Here we use an ad hoc but more elementary method. The twisted Hilbert space has the form

$$\mathcal{H} = \oplus_j V_j \otimes \overline{V_{k/2-j}} . \tag{3.5}$$

Therefore, the only possible Ishibashi state which can appear in this twisted Cardy state is $|\frac{k}{4}\rangle\rangle_{\text{tw}}$:

$$|\frac{k}{4}\rangle_{\text{tw}} \propto |\frac{k}{4}\rangle\rangle_{\text{tw}}. \tag{3.6}$$

Let us note that we have

$$\langle\frac{k}{4}|e^{-2\pi H/\delta}|\frac{k}{4}\rangle = \chi_0(i\delta) + \chi_1(i\delta) + \chi_2(i\delta) + \chi_3(i\delta) + \cdots + \chi_{\frac{k}{2}}(i\delta), \tag{3.7}$$

while

$$\langle\langle\frac{k}{4}|e^{-2\pi H/\delta}|\frac{k}{4}\rangle\rangle \propto \chi_0(i\delta) - \chi_1(i\delta) + \chi_2(i\delta) - \chi_3(i\delta) + \cdots + (-1)^{\frac{k}{2}}\chi_{\frac{k}{2}}(i\delta). \tag{3.8}$$

Therefore we should have

$$_{\text{tw}}\langle\frac{k}{4}|e^{-2\pi H/\delta}|\frac{k}{4}\rangle_{\text{tw}} = \chi_0(i\delta) - \chi_1(i\delta) + \chi_2(i\delta) - \chi_3(i\delta) + \cdots + (-1)^{\frac{k}{2}}\chi_{\frac{k}{2}}(i\delta). \tag{3.9}$$

from which we can determine the proportionality coefficient:

$$|\frac{k}{4}\rangle_{\text{tw}} = \left(\frac{k+2}{2}\right)^{\frac{1}{4}} |\frac{k}{4}\rangle\rangle_{\text{tw}}. \tag{3.10}$$

So far we discussed the untwisted and twisted Cardy states in the original diagonal $SU(2)_k$ model. For even $k$, we can orbifold the $\mathbb{Z}_2$ symmetry and then obtain the D-type modular invariants of the $SU(2)_k$ WZW model. Equivalently, for even $k$, we can consider the WZW sigma model whose target space is the group manifold $SO(3)$ rather than $SU(2)$, and the D-type modular invariants are the infrared limit.[13] Denoting the $SU(2)_k$ WZW model as theory A, we can now determine the boundary states of theories D, F and F′ following our general prescription.

1. The boundary states of theory D is obtained by gauging $\mathbb{Z}_2$ (i.e. $\mathbb{Z}_2$ orbifolding). Using

$$\begin{aligned}
|j\rangle^{\text{D}} &= \frac{1}{\sqrt{2}}(|j\rangle^{\text{A}} + |\frac{k}{2} - j\rangle^{\text{A}}), \\
|j\rangle^{\text{D}}_{\text{tw}} &= \frac{1}{\sqrt{2}}(|j\rangle^{\text{A}} - |\frac{k}{2} - j\rangle^{\text{A}}), \quad j = 0, \frac{1}{2}, \ldots, \frac{k-2}{4} \\
|\frac{k}{4}+\rangle^{\text{D}} &= \frac{1}{\sqrt{2}}(|\frac{k}{4}\rangle^{\text{A}} + |\frac{k}{4}\rangle^{\text{A}}_{\text{tw}}), \\
|\frac{k}{4}-\rangle^{\text{D}} &= \frac{1}{\sqrt{2}}(|\frac{k}{4}\rangle^{\text{A}} - |\frac{k}{4}\rangle^{\text{A}}_{\text{tw}}).
\end{aligned} \tag{3.11}$$

These boundary states have been discussed in the context of orbifolding [46].

---

[13]Depending on whether $k = 4n$ and $k = 4n + 2$, these models are called $D_{\text{even}}$ and $D_{\text{odd}}$ models, respectively, and they show rather distinct behaviors, in that the chiral algebra enhances in the former while it does not in the latter. Historically, this made the construction of the boundary states for the latter more difficult, and was initiated in [40, 41]. This paved the way for a more general method applicable to arbitrary simple current orbifolds e.g. in [42–45].

2. The boundary states of theory F is obtained by stacking a Kitaev chain (Arf invariant) before gauging $\mathbb{Z}_2$. Using (2.43),

$$
\begin{aligned}
|j\rangle_{\mathrm{NS}}^{\mathsf{F}} &= (|j\rangle^{\mathsf{A}} + |\tfrac{k}{2} - j\rangle^{\mathsf{A}}) \otimes |+\rangle \,, \\
|j\rangle_{\mathrm{R}}^{\mathsf{F}} &= (|j\rangle^{\mathsf{A}} - |\tfrac{k}{2} - j\rangle^{\mathsf{A}}) \otimes |-\rangle \,, \quad j = 0, \tfrac{1}{2}, ..., \tfrac{k-2}{4} \,, \\
|\tfrac{k}{4}\rangle_{\mathrm{NS}}^{\mathsf{F}} &= |\tfrac{k}{4}\rangle^{\mathsf{A}} \otimes |+\rangle \,, \\
|\tfrac{k}{4}\rangle_{\mathrm{R}}^{\mathsf{F}} &= |\tfrac{k}{4}\rangle_{\mathrm{tw}}^{\mathsf{A}} \otimes |+\rangle \,.
\end{aligned}
\tag{3.12}
$$

3. The boundary states of theory F′ is obtained by stacking a Kitaev chain (Arf invariant) before gauging $\mathbb{Z}_2$, and then further stacking another Kitaev chain. Using (2.44),

$$
\begin{aligned}
|j\rangle_{\mathrm{NS}}^{\mathsf{F}'} &= \frac{1}{\sqrt{2}}(|j\rangle^{\mathsf{A}} + |\tfrac{k}{2} - j\rangle^{\mathsf{A}}) \otimes |+\rangle \,, \\
|j\rangle_{\mathrm{R}}^{\mathsf{F}'} &= \frac{1}{\sqrt{2}}(|j\rangle^{\mathsf{A}} - |\tfrac{k}{2} - j\rangle^{\mathsf{A}}) \otimes |+\rangle \,, \quad j = 0, \tfrac{1}{2}, ..., \tfrac{k-2}{4} \,, \\
|\tfrac{k}{4}\rangle_{\mathrm{NS}}^{\mathsf{F}'} &= \sqrt{2}|\tfrac{k}{4}\rangle^{\mathsf{A}} \otimes |+\rangle \,, \\
|\tfrac{k}{4}\rangle_{\mathrm{R}}^{\mathsf{F}'} &= \sqrt{2}|\tfrac{k}{4}\rangle_{\mathrm{tw}}^{\mathsf{A}} \otimes |-\rangle \,.
\end{aligned}
\tag{3.13}
$$

Before proceeding, we should pause here to mention that the boundary states of the fermionic versions of diagonal unitary minimal models are given essentially by the same formulas, since the primaries $(r,s)$ of the Virasoro minimal models[14] have the fusion rule which is isomorphic to the fusion rule of the $SU(2)$ affine algebra acting separately on $r$ and $s$, and the $\mathbb{Z}_2$ quotient acts only on a single index [47–49]. The resulting boundary states agree with those determined in [25], up to the factor of $\sqrt{2}$ explained at the end of Sec. 2. The $SU(2)_k$ WZW models also have exceptional invariants [50, 51]. We have not studied the boundary states of fermionic versions of these, but again it should be possible to recover them from the discussion of boundary states of fermionic exceptional minimal models in [25].

## 3.2 The Ising model and a massless Majorana fermion

It is well known that the bosonization of a free massless Majorana fermion is the Ising CFT. We will briefly review the boundary states to set up the notation. A careful discussion was given in [52, 53], but our interpretation is slightly different from the one given there.

**Cardy States for the Ising Model:** The Cardy states of the Ising CFT were discussed in [36]. They are $|0\rangle$, $|\tfrac{1}{2}\rangle$ and $|\tfrac{1}{16}\rangle$ and can be written in terms of the Ishibashi states $|0\rangle\!\rangle$, $|\tfrac{1}{2}\rangle\!\rangle$ and $|\tfrac{1}{16}\rangle\!\rangle$

---

[14]The indices $(r,s)$ here are different from those we have used in the previous section to denote the number of $\mathbb{Z}_2$ defects on $T^2$.

via (A.11):

$$|0\rangle = \frac{1}{\sqrt{2}}|0\rangle\rangle + \frac{1}{\sqrt{2}}|\frac{1}{2}\rangle\rangle + \frac{1}{2^{\frac{1}{4}}}|\frac{1}{16}\rangle\rangle ,$$

$$|\frac{1}{2}\rangle = \frac{1}{\sqrt{2}}|0\rangle\rangle + \frac{1}{\sqrt{2}}|\frac{1}{2}\rangle\rangle - \frac{1}{2^{\frac{1}{4}}}|\frac{1}{16}\rangle\rangle , \qquad (3.14)$$

$$|\frac{1}{16}\rangle = |0\rangle\rangle - |\frac{1}{2}\rangle\rangle .$$

Under the $\mathbb{Z}_2$ generator $g$ we have

$$g\,|0\rangle = |\frac{1}{2}\rangle , \quad g\,|\frac{1}{2}\rangle = |0\rangle , \quad g\,|\frac{1}{16}\rangle = |\frac{1}{16}\rangle . \qquad (3.15)$$

Hence $|\frac{1}{16}\rangle$ is invariant under $\mathbb{Z}_2$, while the other two states are exchanged by $\mathbb{Z}_2$. In additional to the three Cardy states, there is also a twisted state $|\frac{1}{16}\rangle_{\mathrm{tw}}$.

**Ishibashi States for the Massless Majorana Fermion:**  For the free Majorana fermion, there are two boundary conditions, $\psi_L = \pm\psi_R$. Following [52], we will denote the boundary states by $|B_\pm\rangle_{NS/R}$. To write down the boundary states, we consider the mode expansion of the fermions: $\psi_L(z) = \sum_r \psi_r z^{-r-\frac{1}{2}}, \psi_R(\bar{z}) = \sum_r \tilde{\psi}_r \bar{z}^{-r-\frac{1}{2}}$ where $r \in \mathbb{Z}$ for the R sector, and $r \in \mathbb{Z} + \frac{1}{2}$ for the NS sector. Note that we have zero modes in the R sector. Boundary Ishibashi states for the free fermion can then be written down by demanding $(\psi_r \mp i\tilde{\psi}_{-r})|B_\pm\rangle\rangle_{NS/R} = 0$ [52]:

$$|B_\pm\rangle\rangle_{NS} = \exp\left( \pm i \sum_{r>0, r\in\mathbb{Z}+\frac{1}{2}} \psi_{-r}\tilde{\psi}_{-r} \right)|0\rangle ,$$

$$|B_\pm\rangle\rangle_R = \exp\left( \pm i \sum_{n>0, n\in\mathbb{Z}} \psi_{-n}\tilde{\psi}_{-n} \right)|\pm_0\rangle , \qquad (3.16)$$

where the sign $\pm$ in the exponent and in the zero mode state $|\pm_0\rangle$ are correlated with the choice of boundary condition $\psi_L = \pm\psi_R$. The vacuum $|0\rangle$ in the NS sector is defined to be annihilated by all the fermion modes with positive index $\psi_r|0\rangle = \tilde{\psi}_r|0\rangle = 0, r > 0$. Let us further specify $|\pm_0\rangle$ in the R sector. There are two fermion zero modes, $\psi_0$ and $\tilde{\psi}_0$. We can now form a complex fermion $c = \psi_0 + i\tilde{\psi}_0$, which then defines a two dimension Hilbert space $|\pm_0\rangle$ via

$$c\,|+_0\rangle = 0 , \quad |-_0\rangle = c^\dagger|+_0\rangle , \quad c^\dagger|-_0\rangle = 0 , \quad \psi_n|\pm_0\rangle = \tilde{\psi}_n|\pm_0\rangle = 0 , \quad \forall n > 0 . \qquad (3.17)$$

To see the fermion parity, note that the state in the NS sector does not have fermion zero modes, and non zero modes come in pairs of left handed and right handed modes. Thus $(-1)^F = 1$ for $|B_\pm\rangle_{NS}$. For the R sector states, the nonzero modes in the exponent come in pairs, hence they do not contribute to nontrivial fermion parity either. However, the zero modes do contribute to nontrivial fermion parity. We declare that the zero mode state $|+_0\rangle$ is even, $(-1)^F|+_0\rangle = |+_0\rangle$, then

$$(-1)^F|-_0\rangle = (-1)^F c^\dagger|+_0\rangle = -c^\dagger(-1)^F|+_0\rangle = -|-_0\rangle , \qquad (3.18)$$

which shows that $|-_0\rangle$ has odd fermion parity. In summary, we have

$$(-1)^F|\pm_0\rangle = \pm|\pm_0\rangle . \qquad (3.19)$$

The fermion parities of various Ishibashi states can now be summarized as follows:

$$(-1)^F = +1: \quad |B_+\rangle\!\rangle_{\text{NS}}, \quad |B_-\rangle\!\rangle_{\text{NS}}, \quad |B_+\rangle\!\rangle_{\text{R}}, \tag{3.20}$$
$$(-1)^F = -1: \quad |B_-\rangle\!\rangle_{\text{R}}.$$

**Cardy States for the Massless Majorana Fermion:** We further construct the Cardy states from the Ishibashi states $|B_\pm\rangle\!\rangle_{\text{NS/R}}$. As discussed in the introduction, we label the boundary states of the fermionic theory with definite quantum numbers. The boundary state is one to one correspondence with the boundary condition labeled by $\pm$, and each boundary state is either in the NS sector or R sector. [15] This means that the boundary states of the free Majorana fermion is proportional to the Ishibashi states, i.e.

$$|B_+\rangle_{\text{NS}} \propto |B_+\rangle\!\rangle_{\text{NS}}, \quad |B_-\rangle_{\text{NS}} \propto |B_-\rangle\!\rangle_{\text{NS}}, \quad |B_+\rangle_{\text{R}} \propto |B_+\rangle\!\rangle_{\text{R}}, \quad |B_-\rangle_{\text{R}} \propto |B_-\rangle\!\rangle_{\text{R}}. \tag{3.21}$$

To determine the proportionality constants in (3.21), one needs to compute the overlap of the Ishibashi states, and demand the boundary states to satisfy the spin Cardy condition (1.9). Since it does not affect the discussion below, we will not work out the normalization constant here. What is important is that the $\mathbb{Z}_2^F$ quantum numbers of the boundary states $|B_\pm\rangle_{\text{NS/R}}$ are the same as those in the Ishibashi states:

$$(-1)^F = +1: \quad |B_+\rangle_{\text{NS}}, \quad |B_-\rangle_{\text{NS}}, \quad |B_+\rangle_{\text{R}}, \tag{3.22}$$
$$(-1)^F = -1: \quad |B_-\rangle_{\text{R}}.$$

By matching the quantum numbers, we find that it is consistent to relate the fermion boundary states $|B_\pm\rangle_{\text{NS/R}}$ and Ising boundary states $|0, \frac{1}{2}, \frac{1}{16}\rangle$ and $|\frac{1}{16}\rangle_{\text{tw}}$ via either (2.43) or (2.44):

$$
\begin{aligned}
|B_-\rangle_{\text{NS}} &= (|0\rangle + |\tfrac{1}{2}\rangle) \otimes |+\rangle, & |B_-\rangle_{\text{NS}} &= \sqrt{2}\,|\tfrac{1}{16}\rangle \otimes |+\rangle, \\
|B_+\rangle_{\text{NS}} &= |\tfrac{1}{16}\rangle \otimes |+\rangle, & |B_+\rangle_{\text{NS}} &= \tfrac{1}{\sqrt{2}}(|0\rangle + |\tfrac{1}{2}\rangle) \otimes |+\rangle, \\
& & \text{or} & \\
|B_-\rangle_{\text{R}} &= (|0\rangle - |\tfrac{1}{2}\rangle) \otimes |-\rangle, & |B_-\rangle_{\text{R}} &= \sqrt{2}\,|\tfrac{1}{16}\rangle_{\text{tw}} \otimes |-\rangle, \\
|B_+\rangle_{\text{R}} &= |\tfrac{1}{16}\rangle_{\text{tw}} \otimes |+\rangle, & |B_+\rangle_{\text{R}} &= \tfrac{1}{\sqrt{2}}(|0\rangle - |\tfrac{1}{2}\rangle) \otimes |+\rangle.
\end{aligned}
\tag{3.23}
$$

These two choices are equally valid. This is because the Ising theory is self-dual under the Krammers-Wannier duality, and therefore when we take A to be the Ising theory we have A $\simeq$ D, which then leads to F $\simeq$ F'. We discuss more details in Sec. 4.3.

### 3.3 $Spin(N)_1$ WZW Model and $N$ Majorana Fermions

We move on to consider $N$ Majorana fermions. As is well-known, this is a fermionization of the $Spin(N)_1$ WZW model. We start by checking this statement.

**The bulk spectrum:** The $Spin(N)_1$ WZW model has three primaries $0$, $v$, $s$ for odd $N$ and four primaries $0$, $v$, $s$, $c$ for even $N$, where the letters specify how the primary transforms under $Spin(N)$: adjoint ($0$), vector ($v$), spinor ($s$), and conjugate spinor ($c$). Their spins are $0$, $\frac{1}{2}$, $\frac{N}{16}$, $\frac{N}{16}$ in this order, and we also denote the primaries by their spins, distinguishing the conjugate

---

[15]This is in contrast with the discussion in [52,53], where they allow the boundary states in the fermionic theory to be a superposition of NS sector and R sector Ishibashi states.

spinor by a tilde if necessary. We consider the diagonal modular invariant as the theory A, and use the $\mathbb{Z}_2$ symmetry generated by the primary $v$. We then have

$$\mathcal{H}_S = V_0 \otimes \overline{V_0} \oplus V_v \otimes \overline{V_v}, \quad \mathcal{H}_U = V_s \otimes \overline{V_s},$$
$$\mathcal{H}_T = V_s \otimes \overline{V_s}, \qquad\qquad \mathcal{H}_V = V_0 \otimes \overline{V_v} \oplus V_v \otimes \overline{V_0} \tag{3.24}$$

for odd $N$ and

$$\mathcal{H}_S = V_0 \otimes \overline{V_0} \oplus V_v \otimes \overline{V_v}, \quad \mathcal{H}_U = V_s \otimes \overline{V_c} \oplus V_c \otimes \overline{V_s},$$
$$\mathcal{H}_T = V_s \otimes \overline{V_s} \oplus V_c \otimes \overline{V_c}, \quad \mathcal{H}_V = V_0 \otimes \overline{V_v} \oplus V_v \otimes \overline{V_0} \tag{3.25}$$

for even $N$.

Now the theory of $N$ Majorana fermions have the Hilbert space

$$\mathcal{H}_{\text{NS}} = (V_0 \oplus V_v) \otimes (\overline{V_0} \oplus \overline{V_v}),$$
$$\mathcal{H}_{\text{R}} = \begin{cases} (V_s \oplus V_c) \otimes (\overline{V_s} \oplus \overline{V_c}) & (N \text{ even}), \\ V_s \otimes \overline{V_s} \oplus V_s \otimes \overline{V_s} & (N \text{ odd}). \end{cases} \tag{3.26}$$

We can now use (1.5) to confirm that this can be identified either with the theory F or F′ for odd $N$. There is a subtlety for even $N$: the standard assignment of $(-1)^F$ in the R sector is to assign $(-1)^F = +1$ for $V_s \otimes \overline{V_s} \oplus V_c \otimes \overline{V_c}$ and $(-1)^F = -1$ for $V_s \otimes \overline{V_c} \oplus V_c \otimes \overline{V_s}$. This is the assignment for the theory F′ but not for F.

**Cardy states of the bosonic model:** Let us now discuss the Cardy states of the bosonic $Spin(N)_1$ WZW model. For odd $N$, there are three primary fields, with conformal weights $(0,0), (\frac{1}{2}, \frac{1}{2}), (\frac{N}{16}, \frac{N}{16})$. The $S$ matrix is the same as the one for the Ising model, hence the three Cardy states have the same expression (3.14) in terms of the Ishibashi states, where we need to replace the label $1/16$ by $N/16$. Under $\mathbb{Z}_2$ we have:

$$g\,|0\rangle = |\tfrac{1}{2}\rangle, \quad g\,|\tfrac{1}{2}\rangle = |0\rangle, \quad g\,|\tfrac{N}{16}\rangle = |\tfrac{N}{16}\rangle, \tag{3.27}$$

in particular $|\frac{N}{16}\rangle$ is $\mathbb{Z}_2$ invariant. This means that we also have a $\mathbb{Z}_2$ twisted sector state $|\frac{N}{16}\rangle_{\text{tw}}$.

For even $N$, the modular $S$ matrix is

$$S = \frac{1}{2} \begin{pmatrix} 1 & 1 & 1 & 1 \\ 1 & e^{i\pi N/4} & -1 & -e^{i\pi N/4} \\ 1 & -1 & 1 & -1 \\ 1 & -e^{i\pi N/4} & -1 & e^{i\pi N/4} \end{pmatrix}, \tag{3.28}$$

where the columns from the left to the right are for the primaries $0, \frac{N}{16}, \frac{1}{2}, \widetilde{\frac{N}{16}}$ in this order. Hence in terms of the Ishibashi states, we have

$$|0\rangle = \frac{1}{\sqrt{2}} \left( |0\rangle\!\rangle + |\tfrac{N}{16}\rangle\!\rangle + |\tfrac{1}{2}\rangle\!\rangle + |\widetilde{\tfrac{N}{16}}\rangle\!\rangle \right),$$
$$|\tfrac{N}{16}\rangle = \frac{1}{\sqrt{2}} \left( |0\rangle\!\rangle + e^{i\pi N/4}|\tfrac{N}{16}\rangle\!\rangle - |\tfrac{1}{2}\rangle\!\rangle - e^{i\pi N/4}|\widetilde{\tfrac{N}{16}}\rangle\!\rangle \right),$$
$$|\tfrac{1}{2}\rangle = \frac{1}{\sqrt{2}} \left( |0\rangle\!\rangle - |\tfrac{N}{16}\rangle\!\rangle + |\tfrac{1}{2}\rangle\!\rangle - |\widetilde{\tfrac{N}{16}}\rangle\!\rangle \right),$$
$$|\widetilde{\tfrac{N}{16}}\rangle = \frac{1}{\sqrt{2}} \left( |0\rangle\!\rangle - e^{i\pi N/4}|\tfrac{N}{16}\rangle\!\rangle - |\tfrac{1}{2}\rangle\!\rangle + e^{i\pi N/4}|\widetilde{\tfrac{N}{16}}\rangle\!\rangle \right). \tag{3.29}$$

Under $\mathbb{Z}_2$, $|0\rangle\!\rangle$ and $|\frac{1}{2}\rangle\!\rangle$ are even, and $|\frac{N}{16}\rangle\!\rangle$ and $|\widetilde{\frac{N}{16}}\rangle\!\rangle$ are odd. In terms of Cardy states we then have

$$g\,|0\rangle = |\frac{1}{2}\rangle\,, \quad g\,|\frac{1}{2}\rangle = |0\rangle\,, \quad g\,|\frac{N}{16}\rangle = |\widetilde{\frac{N}{16}}\rangle\,, \quad g\,|\widetilde{\frac{N}{16}}\rangle = |\frac{N}{16}\rangle\,. \tag{3.30}$$

Hence they form two $\mathbb{Z}_2$ conjugate pairs. There is no $\mathbb{Z}_2$ invariant boundary state, and therefore there are no boundary states in the $\mathbb{Z}_2$ twisted sector.

**Boundary States of free fermions:** We further discuss the Ishibashi states in the fermionic theory. We will preserve the $SO(N)$ global symmetry, hence all flavors of fermions will be in the same sector, and will have the same sign in $\psi^i_+ = \pm\psi^i_-$ for all $i = 1, ..., N$. The Ishibashi states are direct generalization of (3.16),

$$
\begin{aligned}
|B_\pm\rangle\!\rangle^N_{\mathrm{NS}} &= \exp\left( \pm i \sum_{r>0, r\in\mathbb{Z}+\frac{1}{2}} \sum_{j=1}^N \psi^j_{-r}\tilde\psi^j_{-r} \right) |0\rangle\,, \\
|B_\pm\rangle\!\rangle^N_{\mathrm{R}} &= \exp\left( \pm i \sum_{n>0, n\in\mathbb{Z}} \sum_{j=1}^N \psi^j_{-n}\tilde\psi^j_{-n} \right) |\pm_0\rangle\,,
\end{aligned}
\tag{3.31}
$$

where $|0\rangle$ is defined to be annihilated by all $\psi^i_{\mathrm{R}}$'s with $r > 0$, and $|\pm_0\rangle$ is defined as $\otimes_{j=1}^N |\pm^j_0\rangle$ where $|\pm^j_0\rangle$ is determined in (3.17) for the $j$-th flavor. In summary, there are still four Ishibashi states, i.e. $|B_+\rangle\!\rangle^N_{\mathrm{NS}}, |B_-\rangle\!\rangle^N_{\mathrm{NS}}, |B_-\rangle\!\rangle^N_{\mathrm{R}}, |B_+\rangle\!\rangle^N_{\mathrm{R}}$.

Let us analyze the fermion parity for the Ishibashi states. For NS states $|B_\pm\rangle\!\rangle^N_{\mathrm{NS}}$, the non-zero modes do not contribute to fermion parity. For R states $|B_\pm\rangle\!\rangle^N_{\mathrm{R}}$, only the zero mode contributes the fermion parity. The fermion number is the sum of fermion number for each flavor, hence

$$
\begin{cases}
|+^1_0\rangle \otimes \cdots \otimes |+^N_0\rangle\,, & (-1)^F = 1\,, \\
|-^1_0\rangle \otimes \cdots \otimes |-^N_0\rangle\,, & (-1)^F = (-1)^N\,.
\end{cases}
\tag{3.32}
$$

Thus when $N$ is odd, $|B_-\rangle\!\rangle^N_{\mathrm{R}}$ should have $(-1)^F = -1$ and $|B_+\rangle\!\rangle^N_{\mathrm{NS}}, |B_-\rangle\!\rangle^N_{\mathrm{NS}}, |B_-\rangle\!\rangle^N_{\mathrm{R}}$ all have $(-1)^F = 1$. However, when $N$ is even, all Ishibashi states in (3.31) have even fermion parity.

Similar to the discussion in Sec.3.2, the boundary states for $N$ Majorana fermions, denoted as $|B_\pm\rangle^N_{\mathrm{NS/R}}$, are proportional to the corresponding Ishibashi states, $|B_\pm\rangle^N_{\mathrm{NS/R}} \propto |B_\pm\rangle\!\rangle^N_{\mathrm{NS/R}}$. This means that the quantum numbers of the Ishibashi states discussed in the previous paragraph also apply to the boundary states: when $N$ is odd, $|B_-\rangle^N_{\mathrm{R}}$ should have $(-1)^F = -1$ and $|B_+\rangle^N_{\mathrm{NS}}, |B_-\rangle^N_{\mathrm{NS}}, |B_-\rangle^N_{\mathrm{R}}$ all have $(-1)^F = 1$. However, when $N$ is even, all boundary states $|B_\pm\rangle^N_{\mathrm{NS/R}}$ have even fermion parity.

By matching the quantum numbers of the boundary states, we conclude as follows:

1. When $N$ is odd, the boundary states of the fermionic theory are related to those in the bosonic theory in the same way as in $N = 1$ case (3.23).

2. When $N$ is even, all the boundary states satisfy $(-1)^F = 1$. The fermion boundary states $|B_\pm\rangle^N_{\mathrm{NS/R}}$ and the Ising boundary states can only be identified via (2.44):

$$
\begin{aligned}
|B_+\rangle^N_{\mathrm{NS}} &= \frac{1}{\sqrt{2}}(|0\rangle + |\frac{1}{2}\rangle) \otimes |+\rangle\,, & |B_+\rangle^N_{\mathrm{R}} &= \frac{1}{\sqrt{2}}(|0\rangle - |\frac{1}{2}\rangle) \otimes |+\rangle\,, \\
|B_-\rangle^N_{\mathrm{NS}} &= \frac{1}{\sqrt{2}}(|\frac{N}{16}\rangle + |\widetilde{\frac{N}{16}}\rangle) \otimes |+\rangle\,, & |B_-\rangle^N_{\mathrm{R}} &= \frac{1}{\sqrt{2}}(|\frac{N}{16}\rangle - |\widetilde{\frac{N}{16}}\rangle) \otimes |+\rangle\,.
\end{aligned}
\tag{3.33}
$$

To see why (2.43) does not work, we note that all the boundary states in the $Spin(N)_1$ theory are $\mathbb{Z}_2$ breaking states, while (2.43) pairs certain superposition of $\mathbb{Z}_2$ breaking states with $(-1)^F = -1$ fermion boundary states, which is in contradiction with the fact that there are no $(-1)^F = -1$ boundary states in the Majorana fermion theory. This means that if $Spin(N)_1$ WZW is theory A, then $N$ massless Majorana fermions is theory F':

$$N \text{ Majorana Fermions} = \frac{Spin(N)_1 \text{WZW} \times \text{Kitaev}}{\mathbb{Z}_2} \times \text{Kitaev}. \tag{3.34}$$

## 3.4 Maldacena-Ludwig boundary state of eight Majorana fermions

As the last example in this section, let us discuss the boundary state of eight Majorana fermions studied by Maldacena and Ludwig in [54]. This boundary state arises when we study the monopole catalysis of baryon decay [55, 56] and also when we study the Kondo problem [57,58]. It was also recently revisited in [59].

Let us start by recalling the bulk system, which is the special case $N = 8$ of $N$ Majorana fermions we discussed above. In this particular case, the primaries $\frac{1}{2}$, $\frac{N}{16}$ and $\widetilde{\frac{N}{16}}$ all have the same spin. We distinguish them by labeling them as $v$, $s$ and $c$, as commonly done. The $spin(8)_1$ affine algebra has an $S_3$ outer automorphism permuting them. The Maldacena-Ludwig boundary state is characterized by the condition that the left-moving $spin(8)_1$ and the right-moving $spin(8)_1$ are related by the order-2 automorphism $\omega$ exchanging $v \longleftrightarrow s$ and fixing 0 and $c$. We therefore have $J = \omega(\tilde{J})$ at the boundary, and any boundary state $|\phi\rangle$ with such a boundary condition needs to satisfy

$$(J_{-n} + \omega(\tilde{J}_n))|\phi\rangle = 0. \tag{3.35}$$

Note that the boundary conditions and the defect operators discussed up to the previous subsection always used the trivial automorphism to identify the left-moving and right-moving chiral algebras. Note in particular that the $\mathbb{Z}_2$ lines we used repeatedly, generated by the primary $v$, commute with the chiral algebra, and are not to be confused with the automorphisms of $spin(8)_1$ affine algebra.

The property of such $\omega$-twisted boundary states was studied in detail in [60], and we can simply quote the results there. In the $v$-untwisted sector, there are two $\omega$-twisted Ishibashi states

$$|0\rangle\rangle_\omega := R_0(\omega)|0\rangle\rangle, \qquad |c\rangle\rangle_\omega := R_c(\omega)|c\rangle\rangle, \tag{3.36}$$

where $R_0(\omega)$, $R_c(\omega)$ are the representation matrices of $\omega$ on $V_0$ and $V_c$ with the convention that they only act on the holomorphic side. The two $\omega$-twisted Cardy states are then

$$|0\rangle_\omega = |0\rangle\rangle_\omega + |c\rangle\rangle_\omega, \qquad |c\rangle_\omega = |0\rangle\rangle_\omega - |c\rangle\rangle_\omega, \tag{3.37}$$

and they are exchanged by the action of the $\mathbb{Z}_2$ Verlinde line operator $v$.

We then convert this pair of $\mathbb{Z}_2$-breaking states to the theory F' to find the Maldacena-Ludwig boundary state,[16] which is given by

$$|ML\rangle_{NS} = \sqrt{2}|0\rangle\rangle_\omega, \qquad |ML\rangle_R = \sqrt{2}|c\rangle\rangle_\omega. \tag{3.38}$$

---

[16]We were lucky that the $\omega$-twisted boundary states on the bosonic side turned out to be $\mathbb{Z}_2$-breaking. If it were $\mathbb{Z}_2$-preserving, we would have also needed the $\omega$-twisted boundary state in the $v$-twisted sector. The general theory of such 'doubly-twisted' boundary states do not seem to be readily available.

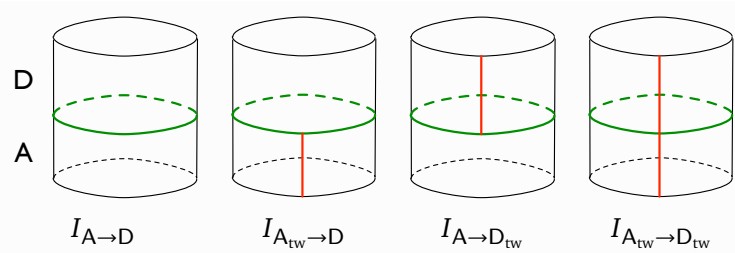

Figure 2: Interfaces from the theory A to D.

We can then find, for example,

$$
\begin{aligned}
{}_{\mathrm{NS}}\langle \mathrm{ML}|e^{-2\pi H/\delta}|\mathrm{ML}\rangle_{\mathrm{NS}} &= \chi_0(i\delta) + \chi_v(i\delta) + \chi_s(i\delta) + \chi_c(i\delta), \\
{}_{\mathrm{R}}\langle \mathrm{ML}|e^{-2\pi H/\delta}|\mathrm{ML}\rangle_{\mathrm{R}} &= \chi_0(i\delta) - \chi_v(i\delta) - \chi_s(i\delta) + \chi_c(i\delta).
\end{aligned}
\tag{3.39}
$$

# 4 Interfaces and boundary states

In the last section we derived the form of the boundary states of fermionic theories F and F$'$. In this section we use this knowledge to determine the algebra of the interfaces between the theories A, D, F and F$'$. This analysis also allows us to see how the anomalous chiral $\mathbb{Z}_2$ symmetry arises in the theory F $\simeq$ F$'$ when A $\simeq$ D.

## 4.1 A $\leftrightarrow$ D

We start by considering the interface between the theory A and D. When placed on a circle, such an interface determines various operators, some of which are shown in Fig 2. Namely, from A to D we have

$$
\begin{aligned}
I_{\mathrm{A}\to\mathrm{D}} &: \mathcal{H}^{\mathrm{A}} \to \mathcal{H}^{\mathrm{D}}, & I_{\mathrm{A}\to\mathrm{D}_{\mathrm{tw}}} &: \mathcal{H}^{\mathrm{A}} \to \mathcal{H}^{\mathrm{D}}_{\mathrm{tw}}, \\
I_{\mathrm{A}_{\mathrm{tw}}\to\mathrm{D}} &: \mathcal{H}^{\mathrm{A}}_{\mathrm{tw}} \to \mathcal{H}^{\mathrm{D}}, & I_{\mathrm{A}_{\mathrm{tw}}\to\mathrm{D}_{\mathrm{tw}}} &: \mathcal{H}^{\mathrm{A}}_{\mathrm{tw}} \to \mathcal{H}^{\mathrm{D}}_{\mathrm{tw}}.
\end{aligned}
\tag{4.1}
$$

The interface operators from D to A can be considered in a similar manner and they are adjoints of the operators given in (4.1).

From the known relations among the Hilbert spaces (1.5) of A and D, we know that

$$
I_{\mathrm{A}\to\mathrm{D}} \propto P_S, \quad I_{\mathrm{A}\to\mathrm{D}_{\mathrm{tw}}} \propto P_T, \quad I_{\mathrm{A}_{\mathrm{tw}}\to\mathrm{D}} \propto P_U, \quad I_{\mathrm{A}_{\mathrm{tw}}\to\mathrm{D}_{\mathrm{tw}}} \propto P_V,
\tag{4.2}
$$

where $P_{S,T,U,V}$ are the projections to the respective components in (1.5).

For example, we have $\mathcal{H}^{\mathrm{A}} = \mathcal{H}_S \oplus \mathcal{H}_T$ and $\mathcal{H}^{\mathrm{D}} = \mathcal{H}_S \oplus \mathcal{H}_T$. Therefore, $I_{\mathrm{A}\to\mathrm{D}}$ can only map the states in $\mathcal{H}_S \subset \mathcal{H}^{\mathrm{A}}$ to $\mathcal{H}_S \subset \mathcal{H}^{\mathrm{D}}$. We expect that $I_{\mathrm{D}\to\mathrm{A}}I_{\mathrm{A}\to\mathrm{D}}$ is proportional to the identity on $\mathcal{H}_S$. The two copies of $\mathcal{H}_S$ belong to two distinct Hilbert spaces[17] $\mathcal{H}^{\mathrm{A}}$ and $\mathcal{H}^{\mathrm{D}}$, so we use an appropriate multiple of $I_{\mathrm{D}\leftrightarrow\mathrm{A}}$ to identify them, resulting in the simple equations (4.2).

The proportionality coefficients can be fixed in various ways. Here, we require that the application of the interface operators to the boundary states give rise to an integral linear

---

[17]When A $\simeq$ D, $I_{\mathrm{A}\to\mathrm{D}}$ can define a non-trivial order-2 operation on $\mathcal{H}_S$. We will discuss this possibility in more detail in Sec. 4.3.

Table 1: Action of interface operators (between theories A and D) on the boundary states.

|  | $|i\rangle^A$ | $|i\rangle^A_{tw}$ | $|a\pm\rangle^A$ |
|---|---|---|---|
| $I_{A\to D}$ | $|i+\rangle^D + |i-\rangle^D$ | - | $|a\rangle^D$ |
| $I_{A\to D_{tw}}$ | $0$ | - | $\pm|a\rangle^D_{tw}$ |
| $I_{A_{tw}\to D}$ | - | $|i+\rangle^D - |i-\rangle^D$ | - |
| $I_{A_{tw}\to D_{tw}}$ | - | $0$ | - |
|  | $|a\rangle^D$ | $|a\rangle^D_{tw}$ | $|i\pm\rangle^D$ |
| $I_{D\to A}$ | $|a+\rangle^A + |a-\rangle^A$ | - | $|i\rangle^A$ |
| $I_{D_{tw}\to A}$ | - | $|a+\rangle^A - |a-\rangle^A$ | - |
| $I_{D\to A_{tw}}$ | $0$ | - | $\pm|i\rangle^A_{tw}$ |
| $I_{D_{tw}\to A_{tw}}$ | - | $0$ | - |

combination of the boundary states. Comparing (1.13) and (1.14), we find[18]

$$I_{A\to D} = \sqrt{2}P_S, \quad I_{A\to D_{tw}} = \sqrt{2}P_T, \quad I_{A_{tw}\to D} = \sqrt{2}P_U, \quad I_{A_{tw}\to D_{tw}} = \sqrt{2}P_V. \tag{4.3}$$

and similarly for the interfaces from D to A. We then find, for example,

$$I_{A\to D}|i\rangle^A = |i+\rangle^D + |i-\rangle^D, \qquad I_{A\to D}|a\pm\rangle^A = |a\rangle^D \tag{4.4}$$

and

$$I_{D\to A}I_{A\to D} = 1 + g, \tag{4.5}$$

where $g$ is the $\mathbb{Z}_2$ generator. Here the right hand side is restricted to act on the untwisted sector. This last relation is known to generalize to

$$I_{D\to A}I_{A\to D} = \sum_{g\in G} g, \tag{4.6}$$

when A is $G$-symmetric and D is the $G$-gauged theory [61]. Systematically, we tabulate the action of interface operators between A and D on the boundary states in table 1.

## 4.2 A ↔ F, F′

Next we consider the interfaces between the original theory and the fermionized versions. We first consider the ones between A and F′. We denote the interfaces from A to F′ as

$$I_{A\to F'_{NS}} : \mathcal{H}^A \to \mathcal{H}^{F'}_{NS}, \qquad I_{A\to F'_R} : \mathcal{H}^A \to \mathcal{H}^{F'}_R,$$
$$I_{A_{tw}\to F'_{NS}} : \mathcal{H}^A_{tw} \to \mathcal{H}^{F'}_{NS}, \quad I_{A_{tw}\to F'_R} : \mathcal{H}^A_{tw} \to \mathcal{H}^{F'}_R. \tag{4.7}$$

Again they are proportional to the projectors $P_{S,T,U,V}$:

$$I_{A\to F'_{NS}} \propto P_S, \quad I_{A\to F'_R} \propto P_T, \quad I_{A_{tw}\to F'_{NS}} \propto P_V, \quad I_{A_{tw}\to F'_R} \propto P_U. \tag{4.8}$$

The proportionality coefficients can be found by studying their actions on boundary states. We find that

$$I_{A\to F'_{NS}} = \sqrt{2}P_S, \quad I_{A\to F'_R} = \sqrt{2}P_T, \quad I_{A_{tw}\to F'_{NS}} = \sqrt{2}P_V, \quad I_{A_{tw}\to F'_R} = \sqrt{2}P_U. \tag{4.9}$$

---

[18]Strictly speaking this method does not determine $I_{A_{tw}\to D_{tw}}$, since the projection of the boundary states is zero.

Table 2: Action of interface operators (between theories A and F′) on the boundary states.

| | $|i\rangle^A$ | $|i\rangle^A_{tw}$ | $|a+\rangle^A$ | $|a-\rangle^A$ |
|---|---|---|---|---|
| $I_{A\to F'_{NS}}$ | $|i\psi\rangle^{F'}_{NS}$ | - | $|a\rangle^{F'}_{NS}$ | $|a\rangle^{F'}_{NS}$ |
| $I_{A\to F'_R}$ | 0 | - | $+|a\rangle^{F'}_R$ | $-|a\rangle^{F'}_R$ |
| $I_{A_{tw}\to F'_{NS}}$ | - | 0 | - | - |
| $I_{A_{tw}\to F'_R}$ | - | $|i\psi\rangle^{F'}_R$ | - | - |
| | $|i\psi\rangle^{F'}_{NS}$ | $|i\psi\rangle^{F'}_R$ | $|a\rangle^{F'}_{NS}$ | $|a\rangle^{F'}_R$ |
| $I_{F'_{NS}\to A}$ | $2|i\rangle^A$ | - | $|a+\rangle^A + |a-\rangle^A$ | - |
| $I_{F'_R\to A}$ | - | 0 | - | $|a+\rangle^A - |a-\rangle^A$ |
| $I_{F'_{NS}\to A_{tw}}$ | 0 | - | 0 | - |
| $I_{F'_R\to A_{tw}}$ | - | $2|i\rangle^A_{tw}$ | - | 0 |

These interfaces act on the boundary states for example as

$$I_{A\to F'_{NS}}|i\rangle^A = |i\rangle^{F'}_{NS}, \quad I_{A\to F'_{NS}}|a\pm\rangle^A = |a\rangle^{F'}_{NS} \tag{4.10}$$

and

$$I_{F'_{NS}\to A}I_{A\to F'_{NS}} = 1 + g. \tag{4.11}$$

Again the right hand side is restricted to the NS sector only. We tabulate the action of interface operators between A and F′ on the boundary states in table 2. These actions were first determined in [53] for two specific cases when the theory A is the critical Ising model or the tricritical Ising model. They were also discussed in [11].

The interfaces between A and F can be determined in a similar manner. We find

$$I_{A\to F_{NS}} = 2P_S, \quad I_{A\to F_R} = 2P_T, \quad I_{A_{tw}\to F_{NS}} = 2P_V, \quad I_{A_{tw}\to F_R} = 2P_U. \tag{4.12}$$

Note the difference by the factor $\sqrt{2}$ in (4.9) and (4.12). Because of this we find that

$$I_{F_{NS}\to A}I_{A\to F_{NS}} = 2(1 + g). \tag{4.13}$$

We tabulate the action of interface operators between A and F on the boundary states in table 3.

The difference between (4.9) and (4.12) can also be understood by considering the interface between the theories F and F′. Since the only difference between these two theories is the stacking by the nontrivial Kitaev chain, the interface simply hosts an unpaired Majorana mode, and satisfies

$$I_{F_{NS}\to F'_{NS}}I_{F'_{NS}\to F_{NS}} = 2. \tag{4.14}$$

This is the dimension of the Hilbert space generated by two paired Majorana modes.

When acting on to the boundary states, we have

$$I_{F_{NS}\to F'_{NS}}|i\rangle^F_{NS} = |i\rangle^{F'}_{NS}, \quad I_{F_{NS}\to F'_{NS}}|a\rangle^F = 2|a\rangle^{F'}. \tag{4.15}$$

The factor 2 in the second equation comes from the fact that $|a\rangle^F$ already has an unpaired Majorana zero mode, which combines with another Majorana zero mode hosted on the interface between F and F′. We then find

$$I_{F'_{NS}\to F_{NS}}I_{A\to F'_{NS}} = I_{A\to F_{NS}}, \quad I_{F_{NS}\to F'_{NS}}I_{A\to F_{NS}} = 2I_{A\to F'_{NS}}. \tag{4.16}$$

Table 3: Action of interface operators (between theories A and F) on the boundary states.

|  | $|i\rangle^A$ | $|i\rangle^A_{tw}$ | $|a+\rangle^A$ | $|a-\rangle^A$ |
|---|---|---|---|---|
| $I_{A\to F_{NS}}$ | $2|i\rangle^F_{NS}$ | - | $|a\psi\rangle^F_{NS}$ | $|a\psi\rangle^F_{NS}$ |
| $I_{A\to F_R}$ | 0 | - | $+|a\psi\rangle^F_R$ | $-|a\psi\rangle^F_R$ |
| $I_{A_{tw}\to F_{NS}}$ | - | 0 | - | - |
| $I_{A_{tw}\to F_R}$ | - | $2|i\rangle^F_R$ | - | - |
|  | $|i\rangle^F_{NS}$ | $|i\rangle^F_R$ | $|a\psi\rangle^F_{NS}$ | $|a\psi\rangle^F_R$ |
| $I_{F_{NS}\to A}$ | $2|i\rangle^A$ | - | $2(|a+\rangle^A+|a-\rangle^A)$ | - |
| $I_{F_R\to A}$ | - | 0 | - | $2(|a+\rangle^A-|a-\rangle^A)$ |
| $I_{F_{NS}\to A_{tw}}$ | 0 | - | 0 | - |
| $I_{F_R\to A_{tw}}$ | - | $2|i\rangle^A_{tw}$ | - | 0 |

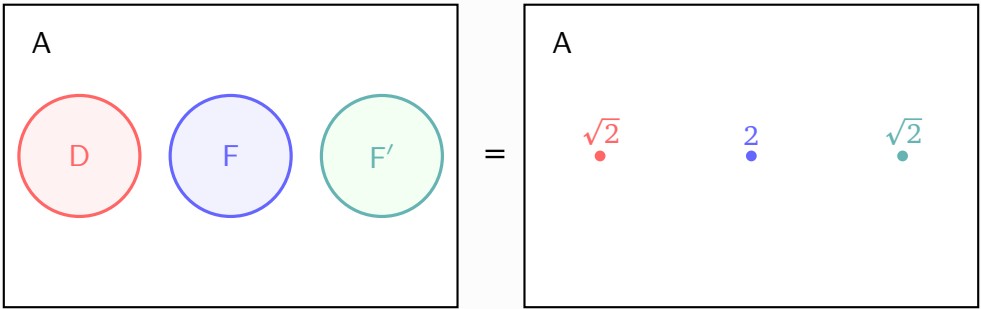

Figure 3: A closed loop of an interface without any operators in it can be evaluated to a number. This number can be called the quantum dimension of the interface, generalizing the same quantity for topological line operators of a single theory.

Finally, we note that the proportionality coefficients

$$I_{A\to D} = \sqrt{2}P_S, \quad I_{A\to F} = 2P_S, \quad I_{A\to F'} = \sqrt{2}P_S, \tag{4.17}$$

translate to the property of the interfaces shown in Fig. 3. Namely, when we have a closed loop of an interface without any operator in it as shown there, they can be simply evaluated to be an insertion of a number, $\sqrt{2}$, 2 or $\sqrt{2}$ depending on whether the interface is from A to D, F or F'. For topological line operators in a single theory, the number obtained by evaluating a closed loop without any operator in it is called the *quantum dimension* of the loop, and here we are dealing with its natural generalization to the interfaces. That the interface between A and F' has a non-integer quantum dimension $\sqrt{2}$ is consistent with the fact that when A is trivial, F' is the nontrivial Kitaev chain.

## 4.3 The special case $A \simeq D$ and the anomalous $\mathbb{Z}_2$ symmetry of $F \simeq F'$

In some important cases, e.g. the Ising model or the tricritical Ising model, the $\mathbb{Z}_2$ orbifold theory D of the original bosonic theory A is equal to itself, $A \simeq D$. In this case the interface $X := I_{A\to D}$ can be considered as a defect of the single theory $A \simeq D$ which now satisfies the

fusion rule

$$X^2 = 1 + g, \qquad gX = X, \tag{4.18}$$

see [62] and also more recent works [27, Sec. 4.3.1] and [29, 63]. We note that, when we wrote in (4.3) above that $I_{A \to D} = \sqrt{2} P_S$, this is meant to be a map obtained by first projecting to the summand $\mathcal{H}_S \subset \mathcal{H}^A$ and then sent isometrically to the corresponding summand $\mathcal{H}_S \subset \mathcal{H}^D$. We now identify $\mathcal{H}^A \simeq \mathcal{H}^D$, but this can introduce a nontrivial unitary operator of order two on $\mathcal{H}_S$. Therefore, the duality interface $X$ as acting on $\mathcal{H}^A$ is a composition

$$X = \sqrt{2} h_S P_S, \tag{4.19}$$

of the projector $P_S$ to $\mathcal{H}_S$ together with a unitary operator

$$h_S : \mathcal{H}_S \to \mathcal{H}_S, \tag{4.20}$$

which squares to 1.

Similarly, the interface $X_{\mathrm{tw}} := I_{A_{\mathrm{tw}} \to D_{\mathrm{tw}}}$ has the form

$$X_{\mathrm{tw}} = \sqrt{2} h_V P_V, \tag{4.21}$$

where

$$h_V : \mathcal{H}_V \to \mathcal{H}_V, \tag{4.22}$$

is again a unitary which squares to one.

The operators $h_S$ and $h_V$ then combine to give a unitary operator

$$h : \mathcal{H}_{\mathrm{NS}}^{\mathsf{F}} \to \mathcal{H}_{\mathrm{NS}}^{\mathsf{F}} \qquad \text{where} \quad \mathcal{H}_{\mathrm{NS}}^{\mathsf{F}} = \mathcal{H}_S \oplus \mathcal{H}_V, \tag{4.23}$$

which squares to one. This is the $\mathbb{Z}_2$ operator acting on the NS sector of the theory $\mathsf{F}$.

For example, in the case of the Ising model treated above, we have $X = \mathcal{L}_{\frac{1}{16}}$, which is the Kramers-Wannier duality defect. Under $X$,

$$X|0\rangle = X|\tfrac{1}{2}\rangle = |\tfrac{1}{16}\rangle, \qquad X|\tfrac{1}{16}\rangle = |0\rangle + |\tfrac{1}{2}\rangle. \tag{4.24}$$

Using (3.23), one finds that the boundary conditions $B_-$ and $B_+$ for the Majorana fermions are exchanged, i.e., $\psi_L = \pm \psi_R$ becomes $\psi_L = \mp \psi_R$, which is precisely the chiral $\mathbb{Z}_2$ symmetry.

On the R-sector, we see that the operator coming from the duality wall $X$ exchanges $\mathcal{H}_U$ and $\mathcal{H}_T$, and in particular flips the sign of $(-1)^F$. This means that the $\mathbb{Z}_2$ symmetry is anomalous. Recall that the anomaly of $\mathbb{Z}_2$ symmetry of fermionic (1+1)-dimensional theory is specified by an integer modulo 8; it is known that the anomalous $\mathbb{Z}_2$ symmetry obtained from a duality defect $X$ satisfying the fusion rule (4.18) automatically has the anomaly characterized by an odd number modulo 8, see e.g. [23, Sec. IV and Sec. VI].

## Acknowledgements

The authors would like to thank the authors of [26] for discussions. The authors would also like to thank Shu-Heng Shao, Philip Boyle Smith, Gerard Watts and an anonymous referee for very helpful comments on the draft. Y.F. thanks discussions and comments of Yuan Yao. He also thanks Shumpei Iino for the collaboration on the work [29] which is closely related to this project. Y.T. is partially supported by JSPS KAKENHI Grant-in-Aid (Wakate-A), No.17H04837

and JSPS KAKENHI Grant-in-Aid (Kiban-S), No.16H06335. Y.T. and Y.Z. are partially supported by WPI Initiative, MEXT, Japan at IPMU, the University of Tokyo. Y.F. acknowledges the support by the QuantiXLie Center of Excellence, a project co-financed by the Croatian Government and European Union through the European Regional Development Fund – the Competitiveness and Cohesion Operational Programme (Grant KK.01.1.1.01.0004).

## A  RCFTs and their $\mathbb{Z}_2$ symmetries

In this section we collect various facts on RCFTs. More specifically, we discuss the diagonal modular invariants and their $\mathbb{Z}_2$ symmetries generated by simple currents of order 2. This will allow us to construct a large class of fermionic models and fermionic boundary states.

**Diagonal modular invariant:**   We start from a chiral algebra $\mathcal{A}$ with irreducible representations $V_\alpha$, where $\alpha = 0, \dots$, such that the label 0 corresponds to the vacuum representation. We denote the fusion rule as

$$V_\alpha V_\beta \sim N^\gamma_{\alpha\beta} V_\gamma. \tag{A.1}$$

The untwisted Hilbert space $\mathcal{H}_0$ has the decomposition

$$\mathcal{H}_0 = \bigoplus_\alpha V_\alpha \otimes \overline{V_\alpha}, \tag{A.2}$$

and therefore has the character

$$\mathrm{Tr}_{\mathcal{H}_0} e^{-2\pi t H} = \sum_\alpha \chi_\alpha(t)\overline{\chi_\alpha(t)}. \tag{A.3}$$

In some references these modular invariants are called charge-conjugation invariants, since the left-mover and the right-movers are complex conjugates of each other.

**Verlinde line operators and generalized twisted sectors:**   In this diagonal theory, we have Verlinde line operators labeled by $\alpha$ [62, 64, 65]. When wrapped around the spatial circle, it determines an operator

$$\mathcal{L}_\alpha : \mathcal{H}_0 \to \mathcal{H}_0, \tag{A.4}$$

satisfying the fusion rule equation

$$\mathcal{L}_\alpha \mathcal{L}_\beta = N^\gamma_{\alpha\beta} \mathcal{L}_\gamma. \tag{A.5}$$

They are known to act in the following manner: by a multiplication by

$$\mathcal{L}_\alpha |\phi_\beta\rangle = \frac{S_{\alpha\beta}}{S_{0\beta}} |\phi_\beta\rangle, \tag{A.6}$$

where $|\phi_\beta\rangle$ is in the summand $V_\beta \otimes \overline{V_\beta}$ of $\mathcal{H}_0$. These equations are compatible because of the celebrated formula of Verlinde [64],

$$N^\gamma_{\alpha\beta} = \sum_\delta \frac{S_{\alpha\delta} S_{\beta\delta} \overline{S_{\gamma\delta}}}{S_{0\delta}}. \tag{A.7}$$

This also means that the Hilbert space $\mathcal{H}_\gamma$ on a circle with an insertion of the Verlinde operator labeled by $\alpha$ is given by

$$\mathcal{H}_\gamma = \bigoplus_{\alpha,\beta} N_{\alpha\gamma}^\beta V_\alpha \otimes \overline{V_\beta}, \tag{A.8}$$

so that the character is

$$\text{Tr}_{\mathcal{H}_\gamma} e^{-2\pi tH} = \sum_{\alpha,\beta} N_{\alpha\gamma}^\beta \chi_\alpha(t)\overline{\chi_\beta(t)}. \tag{A.9}$$

**Ishibashi states and Cardy states:**   To describe the boundary states, we first consider Ishibashi states $|\alpha\rangle\!\rangle \in V_\alpha \otimes \overline{V_\alpha}$ [35] with the property

$$\langle\!\langle\alpha|e^{-2\pi H/\delta}|\beta\rangle\!\rangle = \delta_{\alpha\beta}\chi_\alpha(\frac{i}{\delta}). \tag{A.10}$$

Then the Cardy states are [36]

$$|\alpha\rangle = \sum_\beta \frac{S_{\alpha\beta}}{\sqrt{S_{0\beta}}}|\beta\rangle\!\rangle, \tag{A.11}$$

which satisfies

$$\langle\alpha|e^{-2\pi H/\delta}|\beta\rangle = \text{Tr}_{\mathcal{H}_{\alpha|\beta}} e^{-2\pi\delta H} = \sum_{\alpha,\beta} N_{\alpha\beta}^\gamma \chi_\gamma(i\delta), \tag{A.12}$$

meaning that the open Hilbert space has the decomposition

$$\mathcal{H}_{\alpha|\beta} = \bigoplus_\gamma N_{\alpha\beta}^\gamma V_\gamma. \tag{A.13}$$

We also find the action of the Verlinde operators on the Cardy states as follows:

$$\mathcal{L}_\alpha |\beta\rangle = N_{\alpha\beta}^\gamma |\gamma\rangle. \tag{A.14}$$

**Order-2 simple currents:**   In general, a primary $p$ which has the fusion rule $(V_p)^n \sim V_0$ is called a simple current. The corresponding Verlinde line operator generates a $\mathbb{Z}_n$ symmetry. Here we only consider the case when $n = 2$. Denoting the simple current by $v$, we have the fusion rule $V_v V_v \sim V_0$. The spin $h_v$ is 0 or 1/4 mod 1/2, and the $\mathbb{Z}_2$ is non-anomalous in the former and anomalous in the latter case.

Under the fusion with $v$, primaries can be grouped into two types. Namely, there are those invariant under the multiplication:

$$V_v V_i \sim V_i, \tag{A.15}$$

and pairs exchanged by the multiplication:

$$V_v V_{a+} \sim V_{a-}, \qquad V_v V_{a-} \sim V_{a+}. \tag{A.16}$$

**Cardy states in the twisted sector:**   We now examine the effect of the $\mathbb{Z}_2$ symmetry on the boundary conditions. The equations above translate to the statement that the boundary condition $i$ is invariant under $\mathbb{Z}_2$ while the boundary conditions $a\pm$ break the $\mathbb{Z}_2$ symmetry and are exchanged by it. This means that one can put $\mathbb{Z}_2$-invariant boundary conditions $i, j, \ldots$ on a circle twisted by the $\mathbb{Z}_2$ symmetry. This construction should then determine twisted Cardy states

$$|i\rangle_{\text{tw}} \in \mathcal{H}_v = \bigoplus_\alpha V_\alpha \otimes \overline{V_{v\alpha}}. \tag{A.17}$$

They can be expanded in terms of the twisted Ishibashi states

$$|i\rangle\!\rangle_{\text{tw}} \in V_i \otimes \overline{V}_i \subset \mathcal{H}_v \,, \tag{A.18}$$

having the same normalization as in (A.10). We note that the untwisted and twisted Ishibashi states $|i\rangle\!\rangle$ and $|i\rangle\!\rangle_{\text{tw}}$ labeled by the same symbol $i$ behave in the same way under the action of the chiral algebra $\mathcal{A}$ and its antiholomorphic counterpart $\overline{\mathcal{A}}$, but that they live in different sectors and should better be distinguished.

The expansion of the twisted Cardy states in terms of the twisted Ishibashi states was studied in the case of $\mathbb{Z}_2$ simple currents e.g. in [43]. A general formula applicable for any simple current orbifold was conjectured in [44] which was later proved in [45]. In our case the formula boils down to

$$|i\rangle_{\text{tw}} = \sum_j \frac{\hat{S}_{ij}}{\sqrt{S_{0j}}} |j\rangle\!\rangle_{\text{tw}} \,, \tag{A.19}$$

up to a subtle phase which does not concern us in this paper.[19] Here, $\hat{S}_{ij}$ describes the action of $S \in SL(2,\mathbb{Z})$ on the space of torus conformal blocks with an insertion of the simple current $v$.

The general form of $\hat{S}_{ij}$ was determined e.g. in [66, 67]. They are also known to satisfy a generalized version of the Verlinde formula [67–70]

$$\hat{N}_{i\alpha}^j = \sum_k \frac{\hat{S}_{ik} S_{\alpha k} \overline{\hat{S}_{jk}}}{S_{0k}} \,, \tag{A.20}$$

where the summation is over $\mathbb{Z}_2$ invariant primaries and $\hat{N}_{i\alpha}^j$ is the trace of the $\mathbb{Z}_2$ action $g$ on the fusion space among $i$, $j$ and $\alpha$, see also [46]. This relation guarantees that the overlap of twisted Cardy states satisfies

$$_{\text{tw}}\langle i|e^{-2\pi H/\delta}|j\rangle_{\text{tw}} = \text{Tr}_{\mathcal{H}_{i|j}} g e^{-2\pi \delta H} = \sum_\alpha \hat{N}_{i\alpha}^j \chi_\alpha(i\delta) \,, \tag{A.21}$$

and has a consistent Hamiltonian interpretation in the open channel.

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
