# Peer review of "Fermionization and boundary states in 1+1 dimensions"

_SciPost Physics, doi:SciPost Phys. 11, 082 (2021)_

## Round 1 · Referee Report · Gerard Watts (Referee 1) · 2021-8-3

Report
I would like to thank the author for their reply and for their changes
to the paper which have made it clearer in several places.
First of all, I see now that I was confusing $|a\rangle$ and $|a\rangle\!\rangle$ in (3.23) and the normalisations make sense as they are. Sorry for raising that issue.
Secondly, I am still not happy with the discussion of boundary conditions and boundary states. I am afraid that was not explicit enough in my previous comments, so I will try to make clear what the issue I have is with the definition of boundary states/conditions and the counting of boundary conditions.
It seems to me a fundamental problem that is that the authors have
still not actually defined a boundary condition. The closest they get is the statement above (2.40) that
"The boundary states of $F$ should satisfy the spin Cardy
conditions (1.9). This requires that the coefficients on the right
hand side should be non-negative integers for NS states, while
integers for R states."
I would not argue with this as a constraint, and the states the authors construct appear to satisfy it, but there are more one can add so that the larger set still satisfies this constraint. If a boundary condition $b$ is in one-one correspondence with a pair of states $(|b\rangle_{NS}, |b\rangle_{R})$, and the only constraint on the boundary states is the spin Cardy condition, then the authors have not found the whole set. Following this definition (or constraint), there are actually three boundary conditions for the free fermion. (We found this in our paper [11] in section 5.3.2 but with $|\epsilon\rangle\!\rangle = - |1/2\rangle\!\rangle$ and different conventions to this paper in which the Ramond sector boundary state of a boundary condition with a fermionc zero mode was zero, rather than considering the state with the insertion of an extra zero mode)
In our conventions, these boundary states and their overlaps are
\[
\begin{array}{lll}
\hbox{b.c.} & \phantom{\sqrt 2(}NS & \phantom{-}R \\
\hbox{free } (f) & \sqrt{2}( |0\rangle\!\rangle - |\epsilon\rangle\!\rangle) \qquad& \phantom{-}\
0 \\
\hbox{fixed up }(+) & \phantom{\sqrt 2(}|0\rangle\!\rangle + |\epsilon\rangle\!\rangle & \phantom{-}2^{1/4} |
\sigma \rangle\!\rangle \\
\hbox{fixed down }(-)\qquad & \phantom{\sqrt 2(}|0\rangle\!\rangle + |\epsilon\rangle\!\rangle &\
-2^{1/4} |
\sigma\rangle\!\rangle
\end{array}
\]
These are linearly independent and have overlaps which clearly obey the spin Cardy constraint specified by the authors. In what I hope is obvious notation, these overlaps are
\[
\begin{array}{cclclcl}
{}_{NS} \langle f | e^{-L_1 H} | f \rangle_{NS}
&=& 2(\chi_0(q) + \chi_{1/2}(q))
&=& 2(\chi_0(\tilde q) + \chi_{1/2}(\tilde q))
&=& 2 \chi_{NS}(\tilde q)
\\
{}_{R} \langle f | e^{-L_1 H} | f \rangle_{R}
&=& 0
&&
&=& 0
\\
\\
{}_{NS} \langle f | e^{-L_1 H} | + \rangle_{NS}
&=& \sqrt 2(\chi_0(q) - \chi_{1/2}(q))
&=& 2\chi_{1/16}(\tilde q)
&=& 2\chi_{R}(\tilde q)
\\
{}_{R} \langle f | e^{-L_1 H} | + \rangle_{R}
&=& 0
&&
&=& 0
\\
\\
{}_{NS} \langle f | e^{-L_1 H} | - \rangle_{NS}
&=& \sqrt 2(\chi_0(q) - \chi_{1/2}(q))
&=& 2\chi_{1/16}(\tilde q)
&=& 2\chi_{R}(\tilde q)
\\
{}_{R} \langle f | e^{-L_1 H} | - \rangle_{R}
&=& 0
&&
&=& 0
\\
\\
{}_{NS} \langle + | e^{-L_1 H} | + \rangle_{NS}
&=& \chi_0(q) + \chi_{1/2}(q)
&=& \chi_{0}(\tilde q) + \chi_{1/2}(\tilde q)
&=& \chi_{NS}(\tilde q)
\\
{}_{R} \langle + | e^{-L_1 H} | + \rangle_{R}
&=& \sqrt 2 \chi_{1/16}(q)
&=& \chi_0(\tilde q) - \chi_{1/2}(\tilde q)
&=& \chi_{\widetilde{NS}}(\tilde q)
\\
\\
{}_{NS} \langle + | e^{-L_1 H} | - \rangle_{NS}
&=& \chi_0(q) + \chi_{1/2}(q)
&=& \chi_{0}(\tilde q) + \chi_{1/2}(\tilde q)
&=& \chi_{NS}(\tilde q)
\\
{}_{R} \langle + | e^{-L_1 H} | - \rangle_{R}
&=& -\sqrt 2 \chi_{1/16}(q)
&=& -\chi_0(\tilde q) - \chi_{1/2}(\tilde q)
&=& -\chi_{\widetilde{NS}}(\tilde q)
\\
\\
{}_{NS} \langle - | e^{-L_1 H} | - \rangle_{NS}
&=& \chi_0(q) + \chi_{1/2}(q)
&=& \chi_{0}(\tilde q) + \chi_{1/2}(\tilde q)
&=& \chi_{NS}(\tilde q)
\\
{}_{R} \langle - | e^{-L_1 H} | - \rangle_{R}
&=& \sqrt 2 \chi_{1/16}(q)
&=& \chi_0(\tilde q) - \chi_{1/2}(\tilde q)
&=& \chi_{\widetilde{NS}}(\tilde q)
\end{array}
\]
In terms of the conventions here, these three boundary conditons and
their associated boundary states are
\[
\begin{array}{lll}
\hbox{b.c.} & \phantom{\sqrt 2(}NS & \phantom{-}R \\
B_+ & |1/16\rangle\otimes|+\rangle & \phantom{-(}|1/16\rangle_{tw}\otimes|+\rangle
\\
B_- & (|0\rangle + |1/2\rangle)\otimes|+\rangle & \phantom{-}(|0\rangle - |1/2\rangle)\otimes|-\rangle
\\
B'_- & (|0\rangle + |1/2\rangle)\otimes|+\rangle & -(|0\rangle - |1/2\rangle)\otimes|-\rangle
\end{array}
\]
and their overlaps are
\[
\begin{array}{cclclcl}
{}_{NS} \langle B_+ | e^{-L_1 H} | B_+ \rangle_{NS}
&=& \chi_0(q) + \chi_{1/2}(q)
&=& \chi_{0}(\tilde q) + \chi_{1/2}(\tilde q)
&=& \chi_{NS}(\tilde q)
\\
{}_{R} \langle B_+ | e^{-L_1 H} | B_+ \rangle_{R}
&=& \sqrt 2 \chi_0(q)
&=& \chi_0(\tilde q)-\chi_{1/2}(\tilde q)
&=& \chi_{\widetilde{NS}}(\tilde q)
\\
\\
{}_{NS} \langle B_+ | e^{-L_1 H} | B_- \rangle_{NS}
&=& \sqrt 2(\chi_0(q) - \chi_{1/2}(q))
&=& 2\chi_{1/16}(\tilde q)
&=& 2\chi_{R}(\tilde q)
\\
{}_{R} \langle B_+ | e^{-L_1 H} | B_- \rangle_{R}
&=& 0
&&
&=& 0
\\
\\
{}_{NS} \langle B_+ | e^{-L_1 H} | B'_- \rangle_{NS}
&=& \sqrt 2(\chi_0(q) - \chi_{1/2}(q))
&=& 2\chi_{1/16}(\tilde q)
&=& 2\chi_{R}(\tilde q)
\\
{}_{R} \langle B_+ | e^{-L_1 H} | B'_- \rangle_{R}
&=& 0
&&
&=& 0
\\
\\
{}_{NS} \langle B_- | e^{-L_1 H} | B_- \rangle_{NS}
&=& \chi_0(q) + \chi_{1/2}(q)
&=& \chi_{0}(\tilde q) + \chi_{1/2}(\tilde q)
&=& \chi_{NS}(\tilde q)
\\
{}_{R} \langle B_- | e^{-L_1 H} | B_- \rangle_{R}
&=& \sqrt 2 \chi_{1/16}(q)
&=& \chi_0(\tilde q) - \chi_{1/2}(\tilde q)
&=& \chi_{\widetilde{NS}}(\tilde q)
\\
\\
{}_{NS} \langle B_- | e^{-L_1 H} | B'_- \rangle_{NS}
&=& \chi_0(q) + \chi_{1/2}(q)
&=& \chi_{0}(\tilde q) + \chi_{1/2}(\tilde q)
&=& \chi_{NS}(\tilde q)
\\
{}_{R} \langle B_- | e^{-L_1 H} | B'_- \rangle_{R}
&=& -\sqrt 2 \chi_{1/16}(q)
&=& -\chi_0(\tilde q) - \chi_{1/2}(\tilde q)
&=& -\chi_{\widetilde{NS}}(\tilde q)
\\
\\
{}_{NS} \langle B'_- | e^{-L_1 H} | B'_- \rangle_{NS}
&=& \chi_0(q) + \chi_{1/2}(q)
&=& \chi_{0}(\tilde q) + \chi_{1/2}(\tilde q)
&=& \chi_{NS}(\tilde q)
\\
{}_{R} \langle B'_- | e^{-L_1 H} | B'_- \rangle_{R}
&=& \sqrt 2 \chi_{1/16}(q)
&=& \chi_0(\tilde q) - \chi_{1/2}(\tilde q)
&=& \chi_{\widetilde{NS}}(\tilde q)
\end{array}
\]
These are slightly different because $|f\rangle_R=0$ and $|B_+\rangle_R\neq 0$, but they equally satisfy the spin Cardy constraint.
If the authors have a reason for excluding the (-) or $B'_-$ boundary condition,
please could they explain it?
It is easy enough to formulate some rules which mean that there is not
a one-one correspondence between boundary conditions and pairs of
boundary states, but as fas as I can tell, this is not done in the paper. I am sorry if I missed it.
On the question of the choice of solution for the $P_i$, I stil think one
can do better than the author's comment, but the added comment is ok
and it is much better tha before.
to the paper which have made it clearer in several places.
First of all, I see now that I was confusing $|a\rangle$ and $|a\rangle\!\rangle$ in (3.23) and the normalisations make sense as they are. Sorry for raising that issue.
Secondly, I am still not happy with the discussion of boundary conditions and boundary states. I am afraid that was not explicit enough in my previous comments, so I will try to make clear what the issue I have is with the definition of boundary states/conditions and the counting of boundary conditions.
It seems to me a fundamental problem that is that the authors have
still not actually defined a boundary condition. The closest they get is the statement above (2.40) that
"The boundary states of $F$ should satisfy the spin Cardy
conditions (1.9). This requires that the coefficients on the right
hand side should be non-negative integers for NS states, while
integers for R states."
I would not argue with this as a constraint, and the states the authors construct appear to satisfy it, but there are more one can add so that the larger set still satisfies this constraint. If a boundary condition $b$ is in one-one correspondence with a pair of states $(|b\rangle_{NS}, |b\rangle_{R})$, and the only constraint on the boundary states is the spin Cardy condition, then the authors have not found the whole set. Following this definition (or constraint), there are actually three boundary conditions for the free fermion. (We found this in our paper [11] in section 5.3.2 but with $|\epsilon\rangle\!\rangle = - |1/2\rangle\!\rangle$ and different conventions to this paper in which the Ramond sector boundary state of a boundary condition with a fermionc zero mode was zero, rather than considering the state with the insertion of an extra zero mode)
In our conventions, these boundary states and their overlaps are
\[
\begin{array}{lll}
\hbox{b.c.} & \phantom{\sqrt 2(}NS & \phantom{-}R \\
\hbox{free } (f) & \sqrt{2}( |0\rangle\!\rangle - |\epsilon\rangle\!\rangle) \qquad& \phantom{-}\
0 \\
\hbox{fixed up }(+) & \phantom{\sqrt 2(}|0\rangle\!\rangle + |\epsilon\rangle\!\rangle & \phantom{-}2^{1/4} |
\sigma \rangle\!\rangle \\
\hbox{fixed down }(-)\qquad & \phantom{\sqrt 2(}|0\rangle\!\rangle + |\epsilon\rangle\!\rangle &\
-2^{1/4} |
\sigma\rangle\!\rangle
\end{array}
\]
These are linearly independent and have overlaps which clearly obey the spin Cardy constraint specified by the authors. In what I hope is obvious notation, these overlaps are
\[
\begin{array}{cclclcl}
{}_{NS} \langle f | e^{-L_1 H} | f \rangle_{NS}
&=& 2(\chi_0(q) + \chi_{1/2}(q))
&=& 2(\chi_0(\tilde q) + \chi_{1/2}(\tilde q))
&=& 2 \chi_{NS}(\tilde q)
\\
{}_{R} \langle f | e^{-L_1 H} | f \rangle_{R}
&=& 0
&&
&=& 0
\\
\\
{}_{NS} \langle f | e^{-L_1 H} | + \rangle_{NS}
&=& \sqrt 2(\chi_0(q) - \chi_{1/2}(q))
&=& 2\chi_{1/16}(\tilde q)
&=& 2\chi_{R}(\tilde q)
\\
{}_{R} \langle f | e^{-L_1 H} | + \rangle_{R}
&=& 0
&&
&=& 0
\\
\\
{}_{NS} \langle f | e^{-L_1 H} | - \rangle_{NS}
&=& \sqrt 2(\chi_0(q) - \chi_{1/2}(q))
&=& 2\chi_{1/16}(\tilde q)
&=& 2\chi_{R}(\tilde q)
\\
{}_{R} \langle f | e^{-L_1 H} | - \rangle_{R}
&=& 0
&&
&=& 0
\\
\\
{}_{NS} \langle + | e^{-L_1 H} | + \rangle_{NS}
&=& \chi_0(q) + \chi_{1/2}(q)
&=& \chi_{0}(\tilde q) + \chi_{1/2}(\tilde q)
&=& \chi_{NS}(\tilde q)
\\
{}_{R} \langle + | e^{-L_1 H} | + \rangle_{R}
&=& \sqrt 2 \chi_{1/16}(q)
&=& \chi_0(\tilde q) - \chi_{1/2}(\tilde q)
&=& \chi_{\widetilde{NS}}(\tilde q)
\\
\\
{}_{NS} \langle + | e^{-L_1 H} | - \rangle_{NS}
&=& \chi_0(q) + \chi_{1/2}(q)
&=& \chi_{0}(\tilde q) + \chi_{1/2}(\tilde q)
&=& \chi_{NS}(\tilde q)
\\
{}_{R} \langle + | e^{-L_1 H} | - \rangle_{R}
&=& -\sqrt 2 \chi_{1/16}(q)
&=& -\chi_0(\tilde q) - \chi_{1/2}(\tilde q)
&=& -\chi_{\widetilde{NS}}(\tilde q)
\\
\\
{}_{NS} \langle - | e^{-L_1 H} | - \rangle_{NS}
&=& \chi_0(q) + \chi_{1/2}(q)
&=& \chi_{0}(\tilde q) + \chi_{1/2}(\tilde q)
&=& \chi_{NS}(\tilde q)
\\
{}_{R} \langle - | e^{-L_1 H} | - \rangle_{R}
&=& \sqrt 2 \chi_{1/16}(q)
&=& \chi_0(\tilde q) - \chi_{1/2}(\tilde q)
&=& \chi_{\widetilde{NS}}(\tilde q)
\end{array}
\]
In terms of the conventions here, these three boundary conditons and
their associated boundary states are
\[
\begin{array}{lll}
\hbox{b.c.} & \phantom{\sqrt 2(}NS & \phantom{-}R \\
B_+ & |1/16\rangle\otimes|+\rangle & \phantom{-(}|1/16\rangle_{tw}\otimes|+\rangle
\\
B_- & (|0\rangle + |1/2\rangle)\otimes|+\rangle & \phantom{-}(|0\rangle - |1/2\rangle)\otimes|-\rangle
\\
B'_- & (|0\rangle + |1/2\rangle)\otimes|+\rangle & -(|0\rangle - |1/2\rangle)\otimes|-\rangle
\end{array}
\]
and their overlaps are
\[
\begin{array}{cclclcl}
{}_{NS} \langle B_+ | e^{-L_1 H} | B_+ \rangle_{NS}
&=& \chi_0(q) + \chi_{1/2}(q)
&=& \chi_{0}(\tilde q) + \chi_{1/2}(\tilde q)
&=& \chi_{NS}(\tilde q)
\\
{}_{R} \langle B_+ | e^{-L_1 H} | B_+ \rangle_{R}
&=& \sqrt 2 \chi_0(q)
&=& \chi_0(\tilde q)-\chi_{1/2}(\tilde q)
&=& \chi_{\widetilde{NS}}(\tilde q)
\\
\\
{}_{NS} \langle B_+ | e^{-L_1 H} | B_- \rangle_{NS}
&=& \sqrt 2(\chi_0(q) - \chi_{1/2}(q))
&=& 2\chi_{1/16}(\tilde q)
&=& 2\chi_{R}(\tilde q)
\\
{}_{R} \langle B_+ | e^{-L_1 H} | B_- \rangle_{R}
&=& 0
&&
&=& 0
\\
\\
{}_{NS} \langle B_+ | e^{-L_1 H} | B'_- \rangle_{NS}
&=& \sqrt 2(\chi_0(q) - \chi_{1/2}(q))
&=& 2\chi_{1/16}(\tilde q)
&=& 2\chi_{R}(\tilde q)
\\
{}_{R} \langle B_+ | e^{-L_1 H} | B'_- \rangle_{R}
&=& 0
&&
&=& 0
\\
\\
{}_{NS} \langle B_- | e^{-L_1 H} | B_- \rangle_{NS}
&=& \chi_0(q) + \chi_{1/2}(q)
&=& \chi_{0}(\tilde q) + \chi_{1/2}(\tilde q)
&=& \chi_{NS}(\tilde q)
\\
{}_{R} \langle B_- | e^{-L_1 H} | B_- \rangle_{R}
&=& \sqrt 2 \chi_{1/16}(q)
&=& \chi_0(\tilde q) - \chi_{1/2}(\tilde q)
&=& \chi_{\widetilde{NS}}(\tilde q)
\\
\\
{}_{NS} \langle B_- | e^{-L_1 H} | B'_- \rangle_{NS}
&=& \chi_0(q) + \chi_{1/2}(q)
&=& \chi_{0}(\tilde q) + \chi_{1/2}(\tilde q)
&=& \chi_{NS}(\tilde q)
\\
{}_{R} \langle B_- | e^{-L_1 H} | B'_- \rangle_{R}
&=& -\sqrt 2 \chi_{1/16}(q)
&=& -\chi_0(\tilde q) - \chi_{1/2}(\tilde q)
&=& -\chi_{\widetilde{NS}}(\tilde q)
\\
\\
{}_{NS} \langle B'_- | e^{-L_1 H} | B'_- \rangle_{NS}
&=& \chi_0(q) + \chi_{1/2}(q)
&=& \chi_{0}(\tilde q) + \chi_{1/2}(\tilde q)
&=& \chi_{NS}(\tilde q)
\\
{}_{R} \langle B'_- | e^{-L_1 H} | B'_- \rangle_{R}
&=& \sqrt 2 \chi_{1/16}(q)
&=& \chi_0(\tilde q) - \chi_{1/2}(\tilde q)
&=& \chi_{\widetilde{NS}}(\tilde q)
\end{array}
\]
These are slightly different because $|f\rangle_R=0$ and $|B_+\rangle_R\neq 0$, but they equally satisfy the spin Cardy constraint.
If the authors have a reason for excluding the (-) or $B'_-$ boundary condition,
please could they explain it?
It is easy enough to formulate some rules which mean that there is not
a one-one correspondence between boundary conditions and pairs of
boundary states, but as fas as I can tell, this is not done in the paper. I am sorry if I missed it.
On the question of the choice of solution for the $P_i$, I stil think one
can do better than the author's comment, but the added comment is ok
and it is much better tha before.

Author: Yunqin Zheng on 2021-09-01 [id 1721]
(in reply to Report 1 by Gerard Watts on 2021-08-03)Dear Prof. Watts,
Thank you very much for the extremely detailed explanations. Please find our responses below.
Please note that we define the boundary state in the R-sector of a boundary condition with a boundary Majorana fermion by inserting a Majorana fermion operator on the boundary to make it nonzero. However, if $+\psi$ is a Majorana fermion operator, $-\psi$ is equally good as a Majorana fermion operator (with the same canonical anticommutation relation), but neither sign is privileged over the other. This introduces an intrinsic sign ambiguity in the R-sector boundary state of a boundary condition with a Majorana fermion zero mode. The additional state $|B_-'\rangle_{R}$ mentioned in the reply is simply related to our $|B_-\rangle_{R}$ by this sign ambiguity.
Correspondingly, we added a new equation (2.40) and a footnote 10 in the updated manuscript.
Sincerely,
Yoshiki Fukusumi,
Yuji Tachikawa,
Yunqin Zheng

---

## Round 1 · List of Changes

- Added more discussions on boundary conditions and boundary states in the introduction section.
- Added further justifications to the fermionization maps of the boundary states in section 2.
- Changes which are requested by the referees are implemented.

---

## Round 2 · Referee Report · Gerard Watts (Referee 1) · 2021-9-27

Report
I would again like to thank the authors for their reply.
I do disagree with the authors on the immediate interpretation of the breaking of the sign ambiguity that they have brought about by requiring $P^*_3 P_4 \geq 0$ (does it fix $P_3$ or $P_4$?) but they have now introduced a rationale for reducing the number of boundary states to match their expectations. It could perhaps be that their interpretation is related by duality to mine (although I have not yet worked this out), but I don't see anything will be gained by arguing the point here any more.
As a way to fix the boundary states for boundary conditions which have origins in a physical argument, the spin Cardy constraint seems fine. As a way to define and classify boundary conditions, it seems rather less good - but it is not being used for that here. In the context of this paper, I think it is fine.
So, I would like to thank the authors again for their many improvements and I am happy (despite my reservations) to recommend publication in this form.
I do disagree with the authors on the immediate interpretation of the breaking of the sign ambiguity that they have brought about by requiring $P^*_3 P_4 \geq 0$ (does it fix $P_3$ or $P_4$?) but they have now introduced a rationale for reducing the number of boundary states to match their expectations. It could perhaps be that their interpretation is related by duality to mine (although I have not yet worked this out), but I don't see anything will be gained by arguing the point here any more.
As a way to fix the boundary states for boundary conditions which have origins in a physical argument, the spin Cardy constraint seems fine. As a way to define and classify boundary conditions, it seems rather less good - but it is not being used for that here. In the context of this paper, I think it is fine.
So, I would like to thank the authors again for their many improvements and I am happy (despite my reservations) to recommend publication in this form.

---

## Round 2 · List of Changes

Equation (2.40) and footnote 10 are added.

---

## Editorial Decision

published